# Dissipative preparation of a Floquet topological insulator in an optical lattice via bath engineering

Alexander Schnell[1*], Christof Weitenberg[2,3] and André Eckardt[1]

**1** Technische Universität Berlin, Institut für Theoretische Physik, 10623 Berlin, Germany
**2** IQP Institut für Quantenphysik, Universität Hamburg,
Luruper Chaussee 149, 22761 Hamburg, Germany
**3** The Hamburg Centre for Ultrafast Imaging,
Luruper Chaussee 149, 22761 Hamburg, Germany

* schnell@tu-berlin.de

## Abstract

Floquet engineering is an important tool for realizing topologically nontrivial band structures for charge-neutral atoms in optical lattices. However, the preparation of a topological-band-insulator-type state of fermions, with one nontrivial quasi-energy band filled completely and the others empty, is challenging as a result of both driving induced heating as well as imperfect adiabatic state preparation (with the latter induced by the unavoidable gap closing when passing the topological transition). An alternative procedure that has been proposed is to prepare such states dissipatively, i.e. as a steady state that emerges when coupling the system to reservoirs. Here we discuss a concrete scheme that couples the system to a weakly interacting Bose condensate given by second atomic species acting as a heat bath. Our strategy relies on the engineering of the potential for the bath particles, so that they occupy weakly coupled tubes perpendicular to the two-dimensional system. Using Floquet-Born-Markov theory, we show that the resulting nonequilibrium steady state of the driven-dissipative system approximates a topological insulator. We even find indications for the approximate stabilization of an anomalous Floquet topological insulator, a state that is impossible to realize in equilibrium.



## 1 Introduction

Quantum simulation with ultracold atoms in optical lattices [1–3] has been very successful, especially due to the fact that those systems are generally well isolated from their environment. They therefore allow clean studies of quantum many-body physics, including, for example, the realization of quantum phase transitions [2–9], many-body localization [10–13] and (eigenstate) thermalization [14–18]. Nevertheless, since the atoms are charge neutral, effects that occur in presence of a coupling to external magnetic fields, like (fractional) quantum Hall physics, cannot be studied directly in such systems. A fruitful way of addressing this problem is given by Floquet engineering [19–23], where by using a time-periodic modulation of the Hamiltonian of the system, its dynamics can be modified in such a way that it is described by an effective time-independent Hamiltonian with novel properties [24,25]. In experiments, using this technique, artificial gauge fields have been Floquet engineered [26–30], and (Floquet) bands with nontrivial topology measured [31–35].

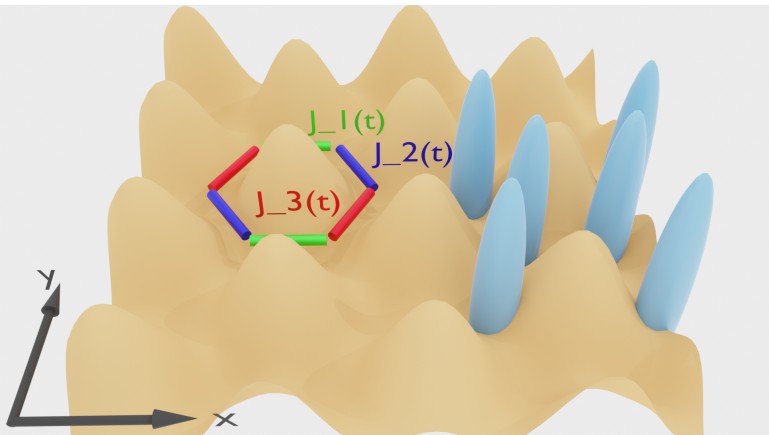

Figure 1: Illustration of the system which is composed of noninteracting fermions in a hexagonal optical lattice with open boundary conditions and $M_x \times M_y$ unit cells. The tunneling matrix elements $J_i(t)$ along the three tunneling directions are modulated periodically with a mutual phase shift of $2\pi/3$. Additionally, every site is coupled to one tube of an array of cigar-shaped bosonic thermal baths.

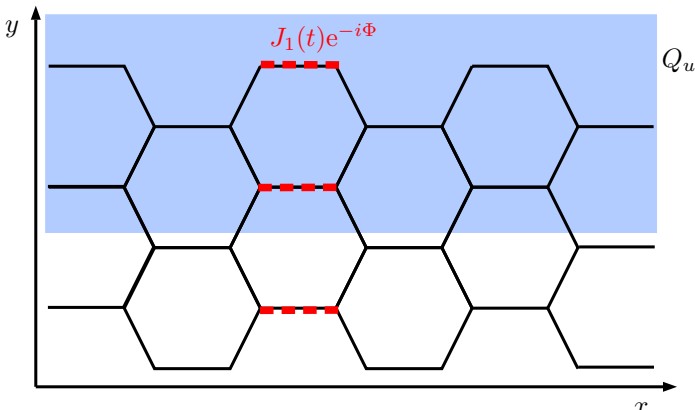

Figure 2: Sketch of the boundary conditions and the charge pump protocol: We add an additional driving phase along the red links at unit cells in the center in the $x$-direction, $l_x = M_x/2$, and then increase $\Phi$ from 0 to $2\pi$ and count charge $Q_u$ in the upper part of the strip.

However, an unwanted side effect of the driving is that the system can and, generically, will be resonantly excited [21, 25, 36–39]. This has been coined Floquet heating and is commonly observed in quantum gas experiments [17, 23, 40–43]. It occurs in interacting systems and limits Floquet engineering to a finite time span (prethermal regime), before heating sets in. However, heating occurs also in noninteracting systems. For instance, as a result of resonant excitations to higher lying bands. In particular for the preparation of integer Chern insulators, moreover, another form of heating occurs. Namely, during the preparation of this state a topological phase transition must be passed, where the band gap between the filled and the neighboring empty band closes, so that particles are excited. Due to the isolated nature of quantum gas systems, both in case of interacting or noninteracting systems, such unwanted excitations will not decay. It was even shown that the adiabatic preparation of topological bands is impossible in the thermodynamic limit [44]. In this paper, we propose the use of a quantum gas mixture as an engineered open quantum system, where one species of atoms forms the Floquet-engineered system, and the second species gives rise to a bath. Similar ideas have been discussed earlier in the context of periodically driven solid-state systems [45–54] or in the case of Floquet topological insulators in contact with generic baths [55–58]. Also explicit criteria under which a Floquet-Gibbs state (i.e. a state that is thermal with respect to the Floquet Hamiltonian) emerges have been discussed [59–63]. However, so far only a few microscopic models for so-engineered baths have been derived, see, e.g., Refs. [64–66]. The results presented in this paper go beyond these studies, since they also discuss how the structuring of the bath can be exploited to engineer an environment that can stabilize the desired Floquet topological state of the driven system.

Quantum gas mixtures with a large number of particles of one species (bath) are a promising platform for the controlled engineering of dissipation in quantum simulators. Such mixtures of two species of quantum gases have gained experimental relevance [67–80], mostly in the context of sympathetic cooling of fermions with a bosonic "buffer gas" for experimental preparation. However, recently there have also been first theoretical [64–66, 81–84] and experimental studies that investigate the *dynamics* of the composite of two ultracold gases. Specifically, there have been successful experimental studies of the impurity relaxation dynamics of a few Caesium-133 atoms in a bath of Rubidium-87 atoms [85–90]. In this setup, one is able to design a species-specific optical lattice and trapping potential [87]. Also, the Rb atoms can be prepared below or above the critical temperature for Bose-Einstein condensation (BEC), thus allowing to tune between a (classical) thermal bath and a BEC bath environment [87].

Motivated by this progress, in this paper, we propose to immerse a fermionic Floquet system, which is a driven two-dimensional (2D) hexagonal lattice of noninteracting spinless fermions, in a bath that consists of a BEC of weakly interacting bosons of a different species, cf. Fig. 1. The system and the bath are assumed to interact weakly, allowing us to use a standard open-quantum-system approach to derive an effective master equation for the reduced system dynamics. We show that in the case where the bath particles are significantly heavier than the system's particles, and where the bath is exposed to the same hexagonal optical lattice as the system, but not confined to a 2D plane, the spectral density of the bath is suitable for our purposes. As will be shown, this allows to prepare effective low-temperature states, which at half filling essentially correspond to a band-insulating ground state. Therefore, starting with any initial state, the system always relaxes to the desired topological-insulator state. Indeed, we find that the so-prepared state gives rise to quantized charge pumping, as expected for a topological Chern insulator. Interestingly, we also find indications of the topological phase transition from this Chern insulator state to a so-called anomalous Floquet topological insulator. The latter is defined by a nontrivial winding number characterizing the state in the spatiotemporal three-dimensional torus given by the product of the first Brillouin zone and one driving period [57, 91]. Accordingly, it cannot be found in non-driven systems.

This paper is organized as follows: In Sec. 2, we introduce the microscopic model of system and bath, and lay out the master equation that governs the reduced dynamics as well as the kinetic equation that describes the dynamics of the driven-dissipative ideal Fermi gas. Sec. 3 discusses the steady states of this kinetic equation, and its effective temperature for the case of $^6$Li in a BEC bath of $^{133}$Cs. For these steady states, in Sec. 4 we present one possibility to perform Laughlin-type charge pumping and show the presence of quantized response in an extensive regime of parameters. Sec. 5 adresses the case of a different spatial configuration of the bath, a three-dimensional (3D) BEC, which occurs naturally when the atoms in the bath are not confined by the hexagonal optical lattice. Also we discuss a different combination of atoms in the mixture, $^{40}$K in $^{87}$Rb. In both cases the effective temperatures of the steady states are higher than in the case of Li in Cs, highlighting the relevance of bath engineering and the importance of a detailed theoretical understanding of such artificial baths. Finally, in Sec. 6 we summarize our results and outline further open questions.

## 2 Model: Modulated optical lattice system and BEC bath

As illustrated in Fig. 1, we investigate noninteracting fermions of mass $m_S$ in a two-dimensional hexagonal optical lattice with $M_x \times M_y$ unit cells and a box shape confinement, leading to open boundary conditions with bearded edge in the $x$-direction and armchair edge in the $y$-direction, as depicted in Fig. 2. The hexagonal lattice is created by optical fields with wavelength $\lambda$, while the transverse confinement into the 2D plane is created using a different wavelength $\lambda_T$. The tunneling elements are time-periodically modulated [34, 35, 92] as we illustrate in Fig. 1. This system is immersed in and interacts weakly with a contact interaction of strength $\gamma$ with an environment of weakly interacting bosons of mass $m_B$ which are subjected to the same optical lattice. The transverse confinement shall be at a wavelength $\lambda_T$ that is a tune-out wavelength of the bosons. Thus, the bosons are only weakly confined in transverse direction (by an additional wide harmonic trap) and live in a hexagonal lattice of one-dimensional tubes, cf. Fig. 1. The tubes are centered at the same lattice sites as the system lattice. We assume that that the temperature $T$ of the bath is low enough such that we can treat the bath as weakly interacting bosons in the superfluid phase. Note that without a trap in the $z$-direction, at finite $T$ and/or for finite interactions bosons are never condensed in one dimension (1D) in the thermodynamic limit. Nevertheless, due to the presence of the wide harmonic trap, there exists

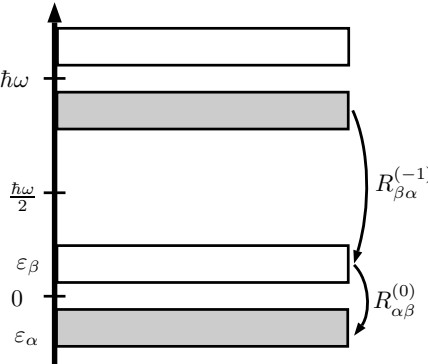

Figure 3: Sketch of the rates in Eq. (11): Since the quasienergies are only defined modulo $\hbar\omega$, there are different kinds of processes that can occur: processes within the same Floquet-Brioullin zone ($K = 0$) or Floquet-Umklapp processes that connect Floquet states of different Brillouin zones ($K \neq 0$).

a crossover temperature to a superfluid phase where the bath is described by the Bogoliubov Hamiltonian and -dispersion [93–95]. Recently it has been studied experimentally, how such 1D tubes of Bose-Einstein condensates absorb excitations [18].

We assume that the mass of the bosons is much larger than the mass of the fermions in the system, $m_B \gg m_S$. Since the tunneling constant between the individual lattice sites for a linear 1D optical lattice scales as $J \approx 4E_R\pi^{-1/2}(V_0/E_R)^{3/4}e^{-2\sqrt{V_0/E_R}}$ [96] with $V_0$ being the lattice depth and recoil energy $E_R = \hbar^2(2\pi/\lambda)^2/(2m)$ (with $m$ being either the system or bath mass), we observe that tunneling in the bath is suppressed exponentially with respect to the square root of the mass when compared to the system, so that in the following we will assume that there is no tunneling between the individual bath tubes. Additionally, since the different atomic species possess different polarizabilities, also the lattice depth $V_0$ will be different for each species. Later we propose a choice of an optical wavelength that leads to larger values of $V_0$ for the bath when compared to the system.

## 2.1 System and bath Hamiltonian

In Bogoliubov approximation (for the bath) and tight-binding approximation with respect to both lattice directions, this model is described by a Holstein Hamiltonian (with vanishing interactions in the system, cf. Appendix B and Refs. [64,97])

$$\hat{H}(t) = \hat{H}_S(t) + \hat{H}_{SB} + \hat{H}_B. \tag{1}$$

The system Hamiltonian describes fermions in a hexagonal lattice with the tunneling matrix elements $J_n(t)$ along the three possible tunneling directions $n = 1, 2, 3$ (see Fig. 1) being modulated periodically in time in a cyclic fashion [34,35,92]

$$\hat{H}_S(t) = -\sum_{\langle \mathbf{l},\mathbf{l'}\rangle} J_{n(\mathbf{l},\mathbf{l'})}(t)\hat{a}_{\mathbf{l}}^\dagger \hat{a}_{\mathbf{l'}}, \tag{2}$$

with $a_{\mathbf{l}}$ being the annihilation operator for a fermionic atom at site $\mathbf{l}$ and $\langle \mathbf{l},\mathbf{l'}\rangle$ denoting a pair of nearest neighbors. The tunneling matrix elements are driven like in the experiment described in Ref. [35]

$$J_n(t) = \frac{J}{2}\left[e^{A\cos(\omega t + \varphi_n)} + 1\right], \tag{3}$$

with dimensionless driving strength $A$, driving frequency $\omega$. The driving phases $\varphi_n = (n-1)2\pi/3$ describe a relative phase lag of $2\pi/3$ between the three different directions, giving rise to a chiral motion that breaks time-reversal symmetry. It is this chiral motion which gives rise to the topologically nontrivial properties of the Floquet-Bloch bands of the system [34, 35, 57, 91, 92, 98–101]. The precise form of driving is, in turn, not relevant and other ways of modulating the tunneling matrix element give rise to similar results [34, 57, 91, 92, 98–101]. For the deeper lattice depth of the bath, the tunnel couplings are practically zero and their modulation does not need to be considered. The cyclic modulation of the lattice-beam intensities has the side-effect of modulated on-site trap frequencies, which are also visible for the bath. Note that this form of driving is favorable compared to lattice shaking [31, 33, 34, 56], where the system would rapidly move relative to the bath.

Since the Hamiltonian of the system is time periodic, $\hat{H}_S(t) = \hat{H}_S(t + \mathcal{T})$ with time period $\mathcal{T} = 2\pi/\omega$, the quasi-steady solutions (Floquet states) $|\psi_\alpha(t)\rangle$ of the Schrödinger equation for the system can be written as $|\psi_\alpha(t)\rangle = \exp(-i\varepsilon_\alpha t/\hbar)|u_\alpha(t)\rangle$ with time-periodic Floquet modes $|u_\alpha(t)\rangle = |u_\alpha(t + \mathcal{T})\rangle$ and quasi energies $\varepsilon_\alpha$ [25]. Note that the quasi energies are only defined up to multiples of $\hbar\omega$ and can, for instance, be restricted to the first Floquet-Brillouin zone $\varepsilon_\alpha \in [-\hbar\omega/2, \hbar\omega/2)$. In this paper, we will most of the time consider the regime of large driving frequencies $\hbar\omega \gg J$, where the two quasi-energy bands of the driven hexagonal lattice together are narrow compared to the width of the Floquet-Brillouin zone (see Fig. 3), indicating that resonant band-coupling processes are suppressed. Such resonant driving induced transitions, where the system exchanges $K$ energy quanta $\hbar\omega$ with the drive, are also known as Floquet-Umklapp processes (in analogy to Umklapp processes where the momentum of reciprocal lattice vectors is provided by a spatially periodic potentials). The Floquet Hamiltonian [21, 25]

$$\hat{H}_{\mathrm{F}} = \frac{i\hbar}{\mathcal{T}} \log \hat{U}(\mathcal{T}, 0) = \sum_\alpha \varepsilon_\alpha |u_\alpha(0)\rangle\langle u_\alpha(0)|, \tag{4}$$

with one-cycle time-evolution operator $\hat{U}(\mathcal{T}, 0)$, is then found to give rise to an effective two-band model for the stroboscopic dynamics at high frequencies [21, 25, 34, 102],

$$\hat{H}_{\mathrm{F}} = -\sum_{\vec{k}} \left(\hat{b}_A^\dagger(\vec{k}), \hat{b}_B^\dagger(\vec{k})\right) \begin{pmatrix} h_1(\vec{k}) & h_0(\vec{k}) \\ h_0(\vec{k})^* & -h_1(\vec{k}) \end{pmatrix} \begin{pmatrix} \hat{b}_A(\vec{k}) \\ \hat{b}_B(\vec{k}) \end{pmatrix}. \tag{5}$$

Here $\hat{b}_{A/B}(\vec{k}) = (M_x M_y)^{-1/2} \sum_{\mathbf{l} \in A/B} \exp(i\vec{k}\vec{r}_\mathbf{l})\hat{a}_\mathbf{l}$ is the annihilation operator at quasimomentum $\hbar\vec{k}$ for sublattice $A, B$ with position $\vec{r}_\mathbf{l}$ of site $\mathbf{l}$. As a consequence of the chiral driving, in the high-frequency limit the two bands of the system are separated by an energy gap and characterized by Chern numbers $+1$ and $-1$ [103], like in the Haldane model [104]. In Fig. 4(a) we show the Chern number $C$ that we obtain numerically (according to Ref. [105]) for the lower quasienergy band of this Floquet Hamiltonian $\hat{H}_{\mathrm{F}}$ for periodic boundary conditions. We observe that at high frequencies, as soon as $A \neq 0$, we have $|C| = 1$. This can be understood also from a Magnus expansion [25, 34, 102], which is presented in Appendix A: From calculating the Chern number for the two leading orders of the high-frequency expansion we find $C = \pm 1$, explaining the topologically nontrivial bands. With increasing driving strengths $A$, in Fig. 4(b), we numerically observe an increase of the effective (direct) quasienergy gap $\Delta_{\mathrm{eff}}$.

At lower frequencies $\omega$ and large driving strength $A$, the system has a topological phase transition to an anomalous Floquet topological band structure. Here, the Chern number vanishes, $C = 0$, as is indicated by the white region in Fig. 4(a). Nevertheless, the system still possesses topologically non-trivial properties, which are associated with the aforementioned spatiotemporal winding numbers [34, 35, 57, 91, 92, 98–101]. This state relies on the resonant coupling between both bands, i.e. on Floquet-Umklapp processes with the system. We will find

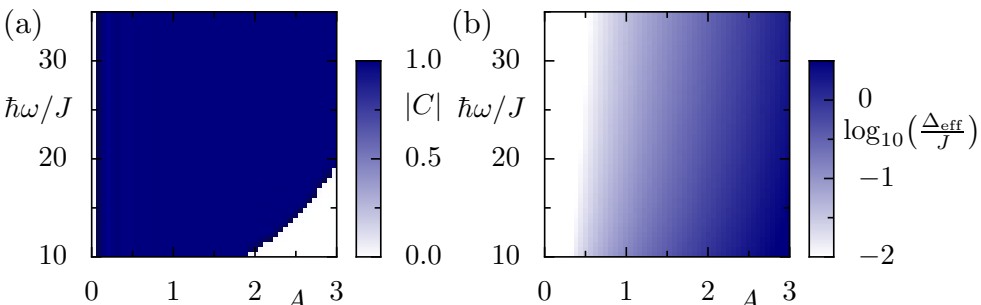

Figure 4: (a) Absolute value of the Chern number $C$ for the lowest band of the Floquet Hamiltonian $\hat{H}_F$, Eq. (2), with *periodic* boundary conditions and $M_x = M_y = 16$ vs. driving strength $A$ and -frequency $\omega$. (b) Value of the effective direct quasienergy gap $\Delta_{eff}$ of $\hat{H}_F$ at the K and K' points, calculated numerically for *periodic* boundary conditions.

indications also of this anomalous Floquet topological phase in the steady state that emerges, when the driven system is coupled to a lattice-trapped bath.

We assume that the bath trap is wide so that each bath tube can be modeled as free 1D Bose gas with periodic boundary conditions in the $z$-direction, which corresponds to a local density approximation at the center of the tube. The bath Hamiltonian then reads (cf. Appendix B for details)

$$\hat{H}_B = \sum_l \sum_q E_B(q) \hat{\beta}_{l,q}^\dagger \hat{\beta}_{l,q} \,, \tag{6}$$

where $l$ labels the individual uncoupled tubes (see Fig. 1) with Bogoliubov quasiparticle annihilation operator $\hat{\beta}_{l,q}$ and where $q$ is the wavenumber in the transverse direction. All tubes have an identical dispersion relation $E_B(q) = \sqrt{E_0(q)^2 + 2GE_0(q)}$ with free dispersion $E_0(q) = \hbar^2 q^2 / 2m_B$ and interaction parameter $G = gn_B$, where $g = 2\pi\hbar^2 a_B/m_B$ is the contact interaction with s-wave scattering length $a_B$ of the bath particles and $n_B = \tilde{n}_B \int dx \int dy |w_0^B(x,y)|^4$ is the volume density at the center of the tube, overlapping with the fermion system. Here $w_0^B$ is the Wannier orbital of the bath lattice and $\tilde{n}_B = N_B/(2M_x M_y L_z)$ the line density, with particle number $N_B$ in the bath, number of lattice sites $2M_x M_y$ and transverse extent $L_z$ of the system. Finally, in leading order of the bath excitations (i.e. only taking into account one-phonon processes) the system–bath Hamiltonian reads (cf. Appendix B for details)

$$\hat{H}_{SB} = \gamma \sum_{l,q} \hat{n}_l \left[ \kappa(q)\hat{\beta}_{l,q} + \kappa(q)^* \hat{\beta}_{l,q}^\dagger \right]. \tag{7}$$

It couples the system's on-site occupations $\hat{n}_l = \hat{a}_l^\dagger \hat{a}_l$ to the respective bath mode $q$ with coupling constants

$$\kappa(q) = c(q) \int d^3r |w_0^S(\vec{r})|^2 |w_0^B(x,y)|^2 e^{iqz} \,. \tag{8}$$

Here $w_0^S(\vec{r})$ denotes the Wannier orbitals of the system and $c(q) = \sqrt{\tilde{n}_B E_0(q)/[L_z E_B(q)]}$.

## 2.2 Floquet-Born-Markov master equation

Within the Bogoliubov approximation, the bath can be viewed as a collection of noninteracting harmonic oscillators describing the quasiparticle excitations. As a result, it is straightforward

to apply the standard Floquet-Born-Markov formalism in order to derive a master equation [63, 106–110]. For sufficiently weak system–bath coupling, the coherences of the density matrix, given by the off-diagonal elements with respect to the Floquet basis, decouple from the diagonal matrix elements and decay, so that we can describe the dynamics of the system using a Pauli type master equation for diagonal elements of the density matrix corresponding to the occupation of the Floquet states. For a single particle, the probability $p_\alpha(t)$ for being in Floquet state $\alpha$ evolves according to

$$\partial_t p_\alpha(t) = \sum_\beta \left( R_{\alpha\beta} p_\beta(t) - R_{\beta\alpha} p_\alpha(t) \right), \tag{9}$$

with single-particle rate [63, 107–110]

$$R_{\alpha\beta} = \sum_{K \in \mathbb{Z}} R_{\alpha\beta}^{(K)}, \tag{10}$$

$$R_{\alpha\beta}^{(K)} = \frac{2\pi\gamma^2}{\hbar} \sum_{\mathrm{l}} |(\nu_{\mathrm{l}})_{\alpha\beta}^{(K)}|^2 g(\varepsilon_\alpha - \varepsilon_\beta + K\hbar\omega), \tag{11}$$

for a jump between Floquet state $\beta$ to $\alpha$ with respective quasienergies $\varepsilon_\beta$ and $\varepsilon_\alpha$ and index $K$ counting the driving quanta $\hbar\omega$ that are exchanged with the bath during the process. Here we have defined the Fourier modes of the coupling matrix elements

$$(\nu_{\mathrm{l}})_{\alpha\beta}^{(K)} = \frac{1}{\mathcal{T}} \int_0^{\mathcal{T}} \mathrm{d}t\, e^{-iK\omega t} \langle u_\alpha(t) | \mathrm{l} \rangle \langle \mathrm{l} | u_\beta(t) \rangle, \tag{12}$$

and the bath-correlation function

$$g(E) = \frac{\mathcal{J}(E)}{e^{E/k_{\mathrm{B}}T} - 1}, \tag{13}$$

where $T$ is the temperature of the bath and $\mathcal{J}(E)$ its spectral density. Note that to obtain the coupling matrix elements $(\nu_{\mathrm{l}})_{\alpha\beta}^{(K)}$ we numerically calculate the Floquet modes $|u_\alpha(t)\rangle$ for *open* boundary conditions rather than the Floquet-Bloch states of the system with periodic boundary conditions.

For the case of many noninteracting fermions in the system, one obtains the equation of motion [111],

$$\partial_t \langle \hat{n}_\alpha \rangle = \sum_\beta \left( R_{\alpha\beta} \langle (1 - \hat{n}_\alpha) \hat{n}_\beta \rangle - R_{\beta\alpha} \langle (1 - \hat{n}_\beta) \hat{n}_\alpha \rangle \right), \tag{14}$$

for the mean occupation $\langle \hat{n}_\alpha \rangle$ of the single-particle Floquet mode $\alpha$. It depends on higher-order particle-particle correlators which leads to a BBGKY-like hierarchy of equations. Here we truncate this hierarchy with the mean-field approximation $\langle \hat{n}_\alpha \hat{n}_\beta \rangle \approx \langle \hat{n}_\alpha \rangle \langle \hat{n}_\beta \rangle$, which leads to the kinetic equations of motion

$$\partial_t \langle \hat{n}_\alpha \rangle = \sum_\beta \left( R_{\alpha\beta} (1 - \langle \hat{n}_\alpha \rangle) \langle \hat{n}_\beta \rangle - R_{\beta\alpha} (1 - \langle \hat{n}_\beta \rangle) \langle \hat{n}_\alpha \rangle \right). \tag{15}$$

These typically yield steady-state distributions that deviate only slightly from exact solutions of the Pauli rate equation for the full many-body Floquet states [111].

Note that two different types of relaxation processes contribute to the rates in Eq. (11), as we sketch in Fig. 3: Processes where $K = 0$ (similar to normal processes in scattering in a crystal lattice) that occur within the same Floquet-Brioullin zone $[-\hbar\omega/2, \hbar\omega/2)$, as well as

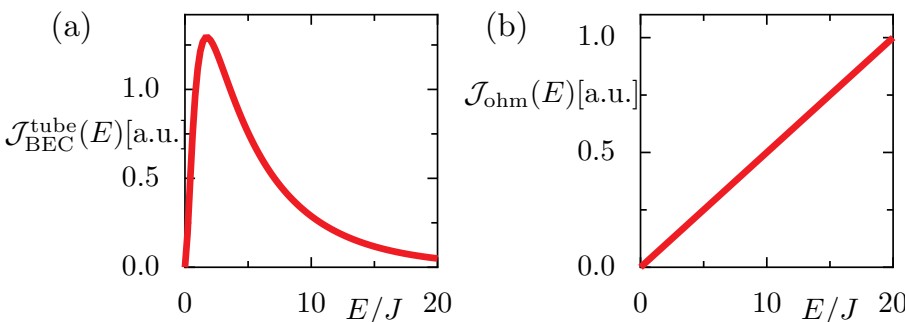

Figure 5: Spectral densities $\mathcal{J}(E)$ as a function of energy $E$ (a) for the engineered lattice-trapped bath sketched in Fig. 1, Eq. (16), and (b) for an ohmic bath, Eq. (17). Parameters for (a) are: $m_S/m_B = 6/133$ (corresponding to $^6$Li in $^{133}$Cs), wavelength of the optical lattice $\lambda = 1064$nm and bath scattering length $a_B = 100a_0$, with Bohr radius $a_0$, corresponding to a Feshbach-tuned scattering length of Cs. Other parameters $n_B = 0.8/\lambda^3$, $d_{T,S} = d_{L,S}$, $V_0/E_R = 8$.

processes where $K \neq 0$ (similar to Umklapp scattering in a crystal lattice) that occur between a state $\beta$ with quasienergy $\varepsilon_\beta$ and the Floquet copy of state $\alpha$ at $\varepsilon_\alpha + K\hbar\omega$. As shown in Fig. 3, those processes are detrimental for an effective thermalization with the temperature of the bath $T$ within the first Brioullin zone, since they involve the exchange of energy with the "drive". This can be viewed also as follows: If only the $K = 0$ processes exist, then the rates $R_{\alpha\beta} = R_{\alpha\beta}^{(0)}$ obey the detailed balance condition $R_{\alpha\beta}/R_{\beta\alpha} = \exp[(\varepsilon_\beta - \varepsilon_\alpha)/(k_B T)]$, which implies thermalization with thermal occupations, where the energies are given by the quasienergies and temperature $T$ [111]. Then at temperatures much lower than the quasienergy gap, $T \ll \Delta_{\text{eff}}$, at half filling the system would form a topological band insulator with the lower band essentially filled completely and the upper one remaining empty (up to excitations that are suppressed exponentially with respect to $\Delta_{\text{eff}}/T$). However, generally, also Floquet Umklapp processes $K \neq 0$ contribute, so the detailed balance condition is broken and the system relaxes to a nonequilibrium steady state. At very low temperatures $T$, the function $g(E)$ in Eq. (11) is nonzero only for processes with $E < 0$. Therefore, even at temperatures $T$ close to zero, processes like $R_{\beta\alpha}^{(-1)}$ in Fig. 3 allow for particles from the Floquet copy of the highly populated lower band to be transferred to the upper band, which leads to an unwanted population increase in the upper Floquet band.

We show in Appendix C that for the BEC-tube bath defined above, one finds

$$\mathcal{J}_{\text{BEC}}^{\text{tube}}(E) = \text{sgn}(E)\frac{n_B}{d^2\pi^2 2}\frac{q(E)}{\sqrt{E^2 + G^2}}e^{-\frac{1}{2}d_{S,T}^2 q(E)^2}. \tag{16}$$

Here $q(E) = \frac{\sqrt{2m_B}}{\hbar}\left(\sqrt{E^2 + G^2} - G\right)^{1/2}$ is the momentum of a Bogoliubov quasiparticle at the transition energy $E$. We, moreover, have introduced the length scale $d = d_B/(1 + d_B^2/d_{S,L}^2)$ where $d_B, d_{S,L}, d_{S,T}$ denote the widths of the Wannier functions for bath particles in lattice direction, system particles in lattice direction, and system particles in transversal direction, respectively. These are defined as harmonic oscillator length $d = \sqrt{\hbar/m\Omega_{\text{eff}}}$, resulting from the approximation of treating the Wannier functions as harmonic oscillator ground states with frequency $\Omega_{\text{eff}}$ in the lattice minima (cf. Appendix C for details). The resulting spectral density is plotted in Fig. 5(a) for an exemplary set of bath parameters, corresponding to $^6$Li in $^{133}$Cs (a mixture which has been successfully prepared experimentally [73–76]). For the choice of $^6$Li and $^{133}$Cs, the interspecies background scattering length is small and negative, but it can be tuned to small positive values via Feshbach resonances [73, 76] to ensure weak system-bath coupling. A possible choice could be the resonance at 843.5G in combination with the

intraspecies Feshbach resonance for $^{133}$Cs at 787G for tuning the bath scattering length to suitable values [112]. For the hexagonal lattice we assume an optical wavelength $\lambda = 1064$nm, which is red-detuned for both atomic species. However, the detuning is larger for $^6$Li than for $^{133}$Cs, which leads to a deeper lattice for the bath atoms as we imagine. We observe that the spectral density in Fig. 5(a) has a rather narrow shape, so that for $\hbar\omega \gg 10J$ rates describing Floquet-Umklapp processes with $K \neq 0$ are suppressed strongly. The narrow width of the spectral density in Eq. (16) is, thus, crucial for the stabilization of the desired band-insulator-type states. It is related to the infinite effective mass of the bath particles in lattice direction as a result of the fact that tunneling of bath particles between different tubes is suppressed (for finite but weak inter-tube tunneling this remains true). The remaining available scattering states are given by the 1D Bogoliubov phonons in the individual tubes. Moreover, the exponential suppression of $\mathcal{J}$ with $q(E)$ is given by the finite width of the Wannier state with respect to momentum. Let us finally remark, that also the collection of tube baths is in total three-dimensional, ensuring that the bath is large compared to the two-dimensional system and, thus, can absorb energy from the system over a long time, before noticeably changing its state.

As a reference, and to show the advantages of the engineered bath of 1D tubes, we will compute steady states also for the case of an ohmic bath, Fig. 5(b), with

$$\mathcal{J}_{\mathrm{ohm}}(E) = E \,. \tag{17}$$

The ohmic bath is a typical choice for a generic phonon bath. It also describes the coupling to an alternative bath environment which is given by a homogeneous 3D ideal gas (bosons or fermions) in the classical limit (at high temperature).

# 3 Steady state distributions and effective temperature

We are now in the position to solve the kinetic equation (15) for the steady-state distributions, $\partial_t \langle \hat{n}_\alpha \rangle = 0$, which describe the nonequilibrium steady state that the system approaches for any initial condition after a relaxation time. For the non-driven system with $A = 0$, we find a Fermi-Dirac distribution with temperature $T$ for the occupation of the single-particle states are given by the eigenstates of the time-independent Hamiltonian $H_{\mathrm{S}}$.

In Fig. 6(a) and (b) we show a typical distribution of the mean occupations $\langle \hat{n}_\alpha \rangle$ of the Floquet modes (red dots) in the driven case, where $A \neq 0$ for (a) the bath of 1D BEC tubes for $^6$Li in $^{133}$Cs and (b) for the ohmic bath for parameters $\hbar\omega = 28.1J, A = 2.8$ marked by the red dot in Fig. 6(c) and (d) respectively. We observe that, even though the system relaxes to a nonequilibrium steady state with nonthermal mean occupations, they still roughly follow a Fermi-Dirac distribution (black dashed line in Fig. 6(a, b))

$$\langle \hat{n}_\alpha \rangle = \frac{1}{e^{(\varepsilon_\alpha - \mu)/(k_{\mathrm{B}} T_{\mathrm{eff}})} + 1} \,, \tag{18}$$

with an effective temperature $T_{\mathrm{eff}}$. Interestingly, we find from our fits that $T_{\mathrm{eff}}$ is in general *not* equal to the temperature $T = 0.01J/k_{\mathrm{B}}$ of the bath, but higher. Here we have chosen the bath temperature $T$ so that it is both small compared to the effective band gap and non-zero, which is favorable for our numerical simulations.

Let us stress that the fact that the distribution $\langle n_\alpha \rangle$ still looks thermal, is nontrivial, since the system relaxes to a genuine nonequilibrium steady state, whose distribution is not dictated by thermodynamics. In Fig. 6(a, b) we observe that the strongest deviation from the thermal distributions is found at energies in the middle of the spectrum close to the gap, where also the largest Berry curvature is located. Hence the topological properties of the state might slightly

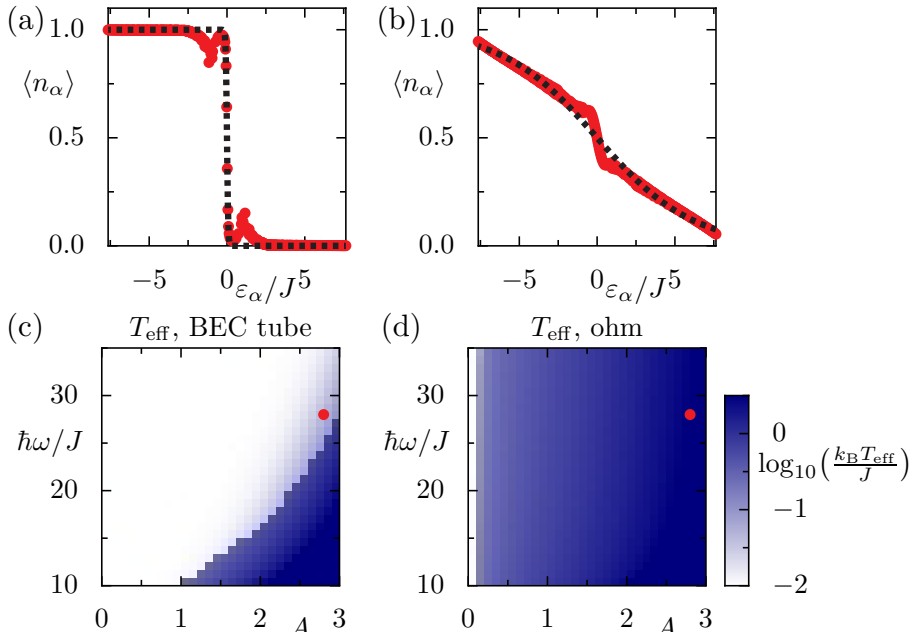

Figure 6: (a, b) Mean occupations $\langle n_\alpha \rangle$ of the Floquet mode $\alpha$ with corresponding quasienergy $\varepsilon_\alpha$ for a system with $M_x = 16$, $M_y = 16$ unit cells of the driven hexagonal lattice model with noninteracting fermions at half filling and open boundary conditions (a) for the BEC bath sketched in Fig. 1, and (b) for an ohmic bath both at $\hbar\omega = 28.1J$, $A = 2.8$ and $k_B T = 0.01J$. The occupations are well described by a Fermi-Dirac distribution (dashed line) with $\mu = 0$ and effective temperature $T_{\text{eff}}$. (c) and (d) show the corresponding effective temperatures, (c) for the BEC bath and (d) for the ohmic bath as a function of driving strength $A$ and -frequency $\omega$. The red dot marks the parameters for (a),(b) respectively.

deviate from a state with corresponding effective temperature. Therefore, we have to perform additional tests. Below we will investigate, whether an approximate quantized response to charge pumping will be found for the steady state.

Due to the finite and rather narrow width of the spectral density of the engineered bath, the effective temperatures $T_{\text{eff}}$ that we determine by fitting a Fermi-Dirac distribution to the mean occupations are much lower for the engineered bath, Fig. 6(c), compared to the ohmic bath, Fig. 6(d). At high frequencies, in a large part of the parameter space the effective temperature $T_{\text{eff}}$ is close or equal to the temperature of the bath $T = 0.01J/k_B$. Thus, we can conclude that the engineered bath of 1D tubes provides a robust means for preparing a Floquet topological band insulator, providing an alternative state preparation scheme that also works in cases where adiabatic preparation of topological bands is hard [44]. Note that in order to stabilize a topological insulating state with one band filled completely, we require that $T_{\text{eff}} \ll \Delta_{\text{eff}}$ (cf. Fig. 4(b)) which is fulfilled in a large parameter regime of the lattice-trapped bath, in stark contrast to the case of a purely ohmic bath where the effective temperature is mostly much higher than the effective gap. In the case of the 2D fermion system embedded in a 3D BEC [64,66], which we discuss later, a mass ratio of $m_S/m_B$ of around 1/100 would be needed to achieve equally low effective temperatures (cf. Sec 5.1). In the next section, we show that the tube-bath scenario indeed gives rise to the quantized response expected for a topological band insulator.

# 4 Charge pumping and quantized response

One hallmark of topological insulators is their quantized response to certain external fields. A typical scenario is given by Laughlin's gedanken experiment. In one variant, a topological insulator on a cylinder is considered through which a magnetic flux $\hbar\Phi/e$ is threaded. Since the energy spectrum will be identical at $\Phi = 0$ and $\Phi = 2\pi$, but the states do not have to reconnect to themselves under a ramping of $\Phi$ from 0 to $2\pi$, an integer number of particles can be transferred, or 'pumped', from the lower boundary of the cylinder to the upper boundary. If only one energy band is completely filled, this number is determined by the Chern number of the band. Hence, in our case, as long as $T_{\text{eff}} \ll \Delta_{\text{eff}}$, with such a protocol we expect the pumping of $Q_{\text{pump}} \approx 1$ in response to the insertion of a flux quantum.

In order to probe this, we consider the configuration depicted in Fig. 2: Along a horizontal line splitting the lattice in half, we modify the tunneling matrix elements in $x$-direction according to $J_1(t) \to J_1(t)\exp(-i\Phi(t))$, where

$$\Phi(t) = 2\pi t/t_{\text{p}}, \tag{19}$$

with $t \in [0, t_{\text{p}}]$. The pumping time $t_{\text{p}}$ should be large compared to $\hbar/\Delta_{\text{eff}}$ where $\Delta_{\text{eff}}$ is the size of the gap around $\varepsilon = 0$. This linear increase of the Peierls phase along the red links can be induced easily by applying an additional on-site potential $\delta = -2\pi\hbar/t_p$ on all lattice sites right of the red colored links in Fig. 2, described by the Hamiltonian

$$\tilde{H}_{\text{S,p}}(t) = -\sum_{\langle \mathbf{l}, \mathbf{l}'\rangle} J_{n(\mathbf{l}, \mathbf{l}')}(t)\hat{a}_{\mathbf{l}}^{\dagger}\hat{a}_{\mathbf{l}'} + \sum_{\mathbf{l}} V_{\mathbf{l}}\hat{n}_{\mathbf{l}}, \tag{20}$$

with $V_{\mathbf{l}} = 0$ for sites $\mathbf{l}$ left of the colored links, and $V_{\mathbf{l}} = \delta$ for sites to the right. Using the gauge transformation $\hat{U}_g(t) = \exp(-i\sum_{\mathbf{l}} V_{\mathbf{l}}\hat{n}_{\mathbf{l}}/\hbar)$ we find the gauge-transformed system Hamiltonian

$$H_{\text{S,p}}(t) = \hat{U}_g(t)^{\dagger}\tilde{H}_{\text{S,p}}(t)\hat{U}_g(t) - i\hbar\hat{U}_g(t)^{\dagger}\dot{\hat{U}}_g(t) = -\sum_{\langle \mathbf{l}, \mathbf{l}'\rangle} J_{n(\mathbf{l}, \mathbf{l}')}(t)e^{-\frac{i}{\hbar}(V_{\mathbf{l}'} - V_{\mathbf{l}})t}\hat{a}_{\mathbf{l}}^{\dagger}\hat{a}_{\mathbf{l}'}. \tag{21}$$

After using the definition of $V_{\mathbf{l}}$ it is apparent that this corresponds to the insertion of the additional phase in Fig. 2. Note also that this gauge transformation leaves the system-bath Hamiltonian $\hat{H}_{\text{SB}}$ invariant. The charge pumping then corresponds to the Hall response to the corresponding force along the colored links. Additionally, we assume that the system is fully relaxed to its nonequilibrium steady state, and that the system-bath coupling is weak enough to be neglected during the ramp.

As a result, during the charge pumping cycle the density matrix for a single particle evolves according to

$$\hat{\varrho}(t) \approx \hat{U}_{\text{S,p}}(t)\hat{\varrho}_{\text{SS}}\hat{U}_{\text{S,p}}(t)^{\dagger}, \tag{22}$$

with time-evolution operator for the system

$$\hat{U}_{\text{S,p}}(t) = \mathcal{T}e^{-\frac{i}{\hbar}\int_0^t \hat{H}_{\text{S,p}}(\tau)d\tau}, \tag{23}$$

and time-ordering operator $\mathcal{T}$. Above, $\hat{\varrho}_{\text{SS}} = \sum_{\alpha} p_{\alpha}^{\text{SS}}|u_{\alpha}(0)\rangle\langle u_{\alpha}(0)|$ denotes the state determined from solving the Pauli rate Eq. (9) for the steady state. In case of many noninteracting fermions, one finds

$$\langle \hat{n}_{\mathbf{l}}\rangle(t) = \sum_{\alpha} |\langle \mathbf{l}|e^{-\frac{i}{\hbar}\int_0^t \hat{H}_{\text{S,p}}(\tau)d\tau}|u_{\alpha}(0)\rangle|^2\langle\hat{n}_{\alpha}\rangle, \tag{24}$$

with $\langle\hat{n}_{\alpha}\rangle$ following from the steady state solution of Eq. (15).

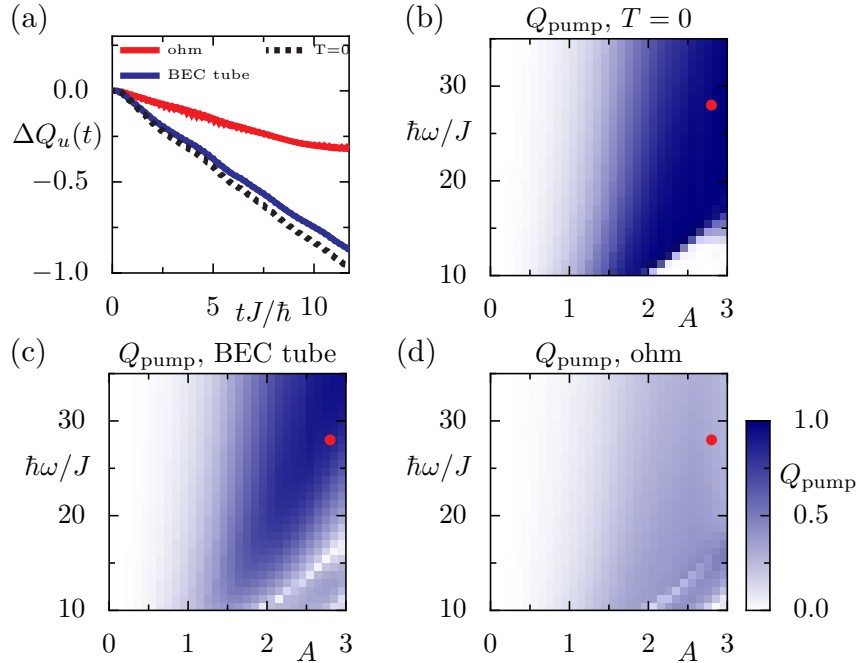

Figure 7: Charge pump: After relaxation to the steady state, we perform the protocol sketched in Fig. 2 in the uncoupled system (we assume that bath relaxation time is much larger than the pump time $t_p = 4.8\hbar/J_{\mathrm{eff}}(A)$). (a) Dynamics of the accumulated charge $\Delta Q_u$ in the upper half of the system as a function of time $t$. (b-d) Accumulated charge $Q_{\mathrm{pump}} = |\Delta Q_u(t_p)|$ at the end of pumping cycle.

We then count the avergage number of particles $Q_u(t)$ in the upper half of the lattice, as shown in Fig. 2, and monitor the pumped charge

$$\Delta Q_u(t) = Q_u(t) - Q_u(0). \tag{25}$$

This leads to curves similar to the one shown Fig. 7(a) again for parameters $\hbar\omega = 28.1J$, $A = 2.8$ for the nonequilibrium steady state with the lattice-trapped bath (blue line), with the ohmic bath (red line) and the ideal case of a Floquet-Gibbs state at $T_{\mathrm{eff}} = 0$ (dashed line). Note that we observe from the numerics, that the pumping time $t_p$ has to be restricted to values $t_p \lesssim 0.15(M_x + M_y)\hbar/J_{\mathrm{eff}}(A) = 4.8\hbar/J_{\mathrm{eff}}(A)$, where we take into account the slowing down of the hopping according to

$$J_{\mathrm{eff}}(A) = \frac{J}{2}\left[1 + I_0(A)\right], \tag{26}$$

the effective tunneling matrix element in the high-frequency approximation of Appendix A and $I_0(z)$ is the modified Bessel function of first kind and 0th order. This is because of finite size effects: For increased values of $t_p$, the edge currents that are excited by the protocol will reach the lower half of the lattice and therefore reduce the charge

$$Q_{\mathrm{pump}} = |\Delta Q_u(t_p)|, \tag{27}$$

that is accumulated after a full cycle of the charge pump.

The resulting values for the charge pump $Q_{\mathrm{pump}}$ are shown in Fig. 7(b) for the ideal case of a Floquet-Gibbs state at $T_{\mathrm{eff}} = 0$, given by

$$\langle n_\alpha \rangle_{T_{\mathrm{eff}}=0} = \Theta(\mu - \varepsilon_\alpha), \tag{28}$$

where at half filling $\mu = 0$, so that only the lower band is filled. Furthermore, we show $Q_{\text{pump}}$ in Fig. 7(c) for the coupling to the tube bath and in Fig. 7(d) for the coupling to an ohmic bath. First of all, we observe that even in the ideal case, Fig. 7(b), due to the finite ramp time required by the finite system size, a near quantized response $Q_{\text{pump}} \approx 1$ is only observed at rather strong driving strengths $A$, where adiabatic pumping is enabled by a sufficiently large gap (see Fig. 4(b)). Second, we observe that for the tube bath the response $Q_{\text{pump}}$ is almost optimal as it coincides largely with the $T_{\text{eff}} = 0$ case. For the ohmic bath Fig. 7(d), however, since the effective temperatures are high when compared to the size of the gap, the protocol does not give rise to quantized charge pumping, with $Q_{\text{pump}}$ on the order of 0.3.

It is intriguing to observe that the transported charge depicted in Fig. 7(d) even allows to identify the topological phase transition, where the Chern number becomes zero [see Figs. 4(a) and 7(b)]. As discussed above, the preparation of effective thermal states is challenged at low frequencies by Floquet-Umklapp processes related to the system–bath coupling. In turn, the phase transition to the anomalous Floquet topological state is connected to Floquet-Umklapp processes associated with resonant band coupling in the system. At the topological phase transition, there is a closing of the gap across the first and second Floquet-Brillouin zone, and the bath allows for population transfer from the Floquet copy of the lower Floquet band to the upper Floquet band (cf. processes with rate $R_{\alpha\beta}^{(-1)}$ in Fig. 3). Therefore, it is not obvious at all, that it is possible to stabilize an approximate anomalous Floquet topological band insulator with a bath. However, the thin white stripe in Fig. 7(c) suggests that there is a small parameter regime, where Umklapp processes inside the system can already give rise to the anomalous Floquet topological band structure, while the Umklapp processes associated with the system-bath coupling are not yet detrimental for the preparation of an approximate band insulator. This is substantiated by Appendix D, where we show visible edge mode dynamics in the anomalous Floquet topological insulator phase in the steady state of the system with the BEC tube bath, albeit at reduced contrast when compared to the ideal $T_{\text{eff}} = 0$ case.

## 5 Variation of the bath parameters

Let us now discuss which features of the proposed lattice-trapped bath are crucial for achieving low $T_{\text{eff}}$. To this end, we discuss in Sec. 5.1 the case with no 1D confinement for the bath, leading to a 3D homogeneous BEC as a bath, as well as in Sec. 5.2 the case of $^{40}\text{K}$ in $^{87}\text{Rb}$ (which has been realized in quantum gas experiments [70, 78, 79]) where the mass ratio is not as imbalanced as for Li in Cs considered so far. In both cases the achievable effective temperatures are too high to observe quantized charge pumping.

### 5.1 Spatially homogeneous BEC bath

As illustrated in Fig. 8, a natural different configuration for the bath is one where the the bath particles do not see the lattice potential of the system, which can be realized by choosing an optical lattice at a tune-out wavelength of the bath. The bath is then given by a continuous 3D weakly interacting BEC. Similar setups have been studied experimentally for one-dimensional undriven lattices as system [80, 87]. Assuming again density-density interactions with weak coupling strength $\gamma$, one finds the spectral density [66]

$$\mathcal{J}_{\text{BEC}}^{\text{3D}}(E) = \text{sgn}(E) \frac{2n_{\text{B}}}{(2\pi)^2 2} \frac{q(E)^3}{\sqrt{E^2 + G^2}} e^{-\frac{1}{2}q(E)^2 d_{\text{S,T}}^2}. \tag{29}$$

Here we also assumed that the bath correlation length is much shorter than the lattice spacing (which is valid if the condition $q(E) \gg 1/\lambda$ is fulfilled at typical transition energies $E$ [66]),

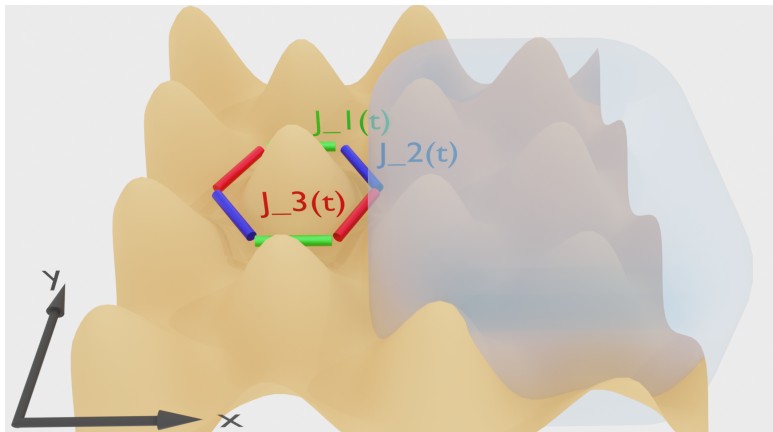

Figure 8: Illustration of the system with 3D bath. Instead of confining the bosons into 1D tubes, we consider a homogeneous 3D BEC as a bath.

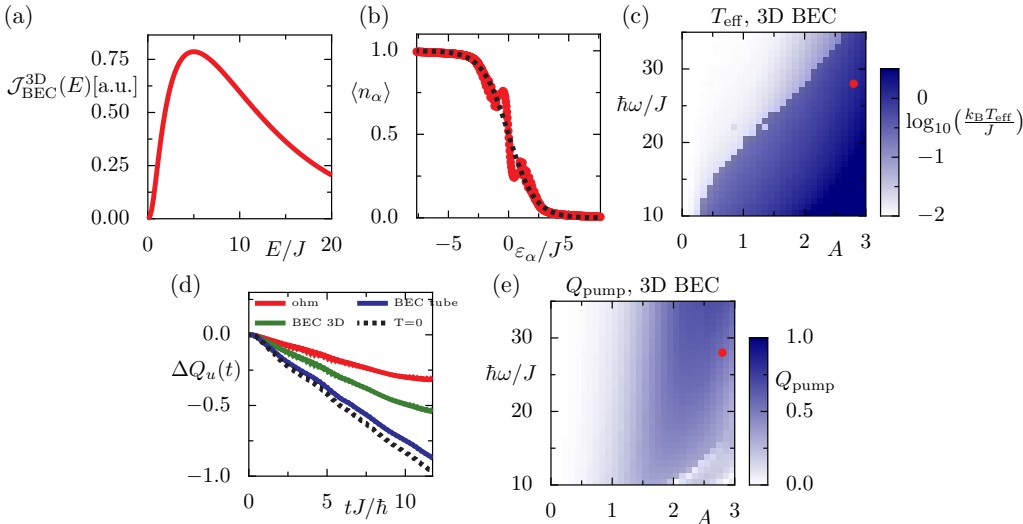

Figure 9: Same as (a) in Fig. 5, (b, c) in Fig. 6, (d, e) in Fig. 7, but without optical lattice for the bath, leading to a 3D homogeneous BEC as a bath. All other parameters are unchanged.

so that the rates are still of the form of Eq. (11). Note that the bath correlation length is set by the correlation length between Bogoliubov quasiparticles, which is finite in spite of the long-range coherence of the underlying BEC [66]. The spectral density for the 3D bath in Eq. (29) is similar to that obtained for the 1D tubes in Eq. (16), however, due to the presence of more accessible bath modes at a given energy $E$, it is not proportional to the wavenumber $q(E)$ anymore, but rather to its cube. This leads to the modified spectral density in Fig. 9(a), which does not drop off as sharply at high energies as for the earlier case in Fig. 5(a), where all other parameters are chosen identically.

As we observe in Fig. 9(b) also in the case of the 3D bath, the distributions $\langle n_\alpha \rangle$ are well described by effective thermal distributions. However, as seen in Fig. 9(c), at given frequency $\omega$, the corresponding effective temperatures $T_{\mathrm{eff}}$ are on the order of $J$ already at much smaller driving strengths $A$, as compared to Fig. 6(c). This is due to the fact that due to the slower decay of the spectral density, Floquet Umklapp processes across multiple Floquet-Briloullin zones are not blocked as efficiently as in the case of the bath consisting of 1D tubes.

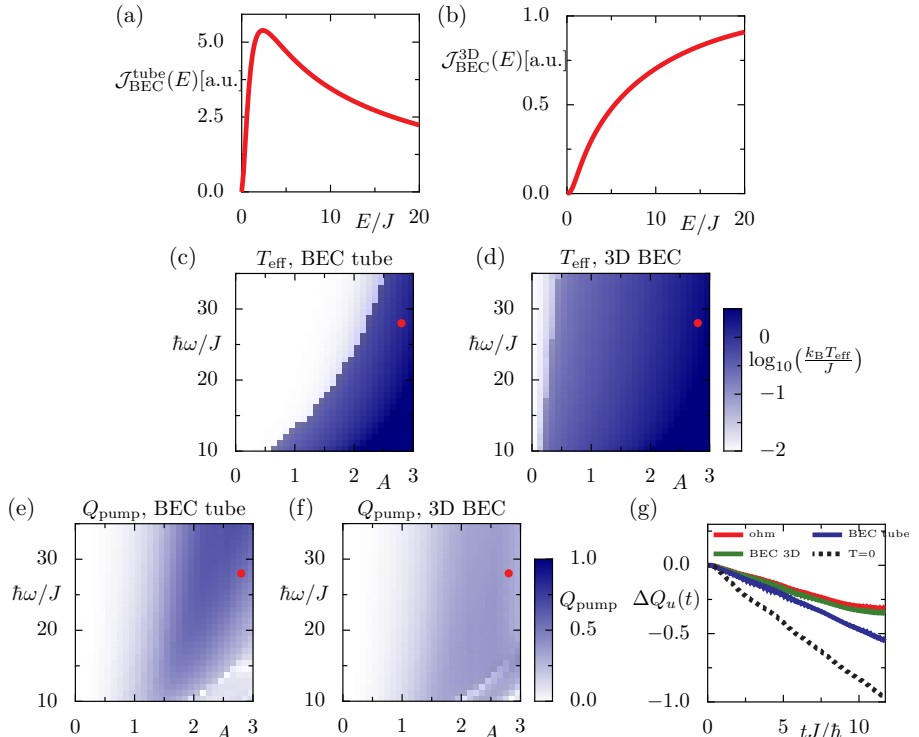

Figure 10: Same as (a, b) in Fig. 5, (c, d) in Fig. 6, (e-g) in Fig. 7, but for $^{40}$K in $^{87}$Rb, $m_S/m_B = 40/87$ both for the case of 1D BEC tube (left panel) as well as the 3D BEC (right panel) as a bath. The bath scattering length of Rubidium is assumed as $a_B = 100a_0$.

Fig. 9(d) and (e) show that, therefore, the achievable charge pumping $Q_{pump}$ is significantly lower than in the case of the 1D tubes, limiting the observable charge transport to the order of 0.5. Although this is an advantage compared to the simple ohmic case, it still shows that for the 3D bath, the Floquet-engineering of a topological insulating phase is not possible due to the high effective temperatures in the steady state, even in the presence of a bath that has seemingly a sufficiently low bath temperature, $T = 0.01J$.

### 5.2 Other atom species combinations

In this section we again study the setup of Fig. 1 with 1D tubes as a bath, as well as the 3D BEC bath in Fig. 8, however we resort to the case of $^{40}$K in a bath of $^{87}$Rb, a mixture which has been prepared in state-of-the-art quantum gas experiments [67,70,72]. The bath scattering length is chosen to the value of $^{87}$Rb, and we again assume a typical optical wavelength $\lambda = 1064$nm. As shown in Fig. 10(a) and (b), the resulting spectral densities for these parameters are much less favorable, as they decay less or not at all with energy. As we observe in Fig. 10(c) and (d), this leads, as expected, to higher effective temperatures in the NESS. The achievable charge pumping, shown in Fig. 10(e), (f) and (g), in the case of the 1D tubes is on the order of 0.6 which is considerably lower than in the almost quantized case for Li in Cs. For K in a 3D Rb BEC, the achievable charge pumping is only on the order of 0.35 and therefore almost as bad as for the generic ohmic bath.

The results of this section clearly highlight the importance of the engineered bath proposed in the previous sections, both regarding the carefully chosen mass ratio and the structuring of the bath as array of 1D tubes. Note, that the mass ratio can also be improved by increasing the effective mass in the bath effectively by switching on a lattice also in the transverse direction.

# 6 Summary and outlook

Floquet engineering is a widely used tool in quantum simulation with ultracold atoms. Nevertheless, heating due to non-adiabatic processes during state preparation and resonant excitations (aka Floquet heating) are often unavoidable. Due to the isolated nature of quantum gas experiments, excitations will remain in the system forever. In this work, we have shown that by coupling a driven system to an engineered lattice-trapped bath given by a Bose condensate of a second atomic species, it is possible to stabilize approximate Gibbs-type states with respect to the Floquet Hamiltonian. As an example we used a system of spinless fermions in a Floquet topological band structure. We demonstrated that an engineered lattice-trapped bath of heavy bosons confined in 1D tubes can stabilize topological insulator states, with one band filled and the other one empty, as steady state. We also showed that (and how) the topological properties of the system can be inferred from quantized topological charge pumping in a large region of parameter space. Remarkably, we also find that it is even possible to observe signatures of the topological phase transition to an anomalous Floquet topological phase, a state that cannot exist in undriven systems.

The proposed dissipative preparation of (anomalous) Floquet topological insulator states has the advantage that the system will relax to the target state from any initial state. This approach is, therefore, very robust. Moreover, it can counteract Floquet heating and it does not suffer from the creation of unwanted excitations, as they can occur during imperfect adiabatic state preparation. The latter is a problem especially for the interesting case of topologically nontrivial states, the preparation of which require passing a topological phase transition, where non-adiabatic processes are induced due to the gap closing at the transition.

An interesting question to be addressed in the future concerns the quantization of other response functions and in particular that of circular dichroism [113, 114]. We expect similar results for such a probe, while, at the same time, the continuous stabilization by an external bath might lead to longer lifetimes than in previous experiments, and thereby facilitate the probing. Besides the transient response, which is limited by having the pumping time to be chosen faster than the relaxation time with the bath, one could also investigate in how far the steady state is changing in response to additional circular driving, when the bath continuously contacts the excitation by the additional drive. Another intriguing question concerns the stabilization of effective thermal states of *interacting* Floquet engineered systems, which, generically are expected to heat up towards infinite temperature [21, 25, 36, 37]. This form of heating is related to resonant excitation processes that happen also in the regime of large frequencies. Recently, it was shown that approximate thermal states of an approximate Floquet Hamiltonian, as it results from a high-frequency approximation, can be stabilized by a thermal bath [59]. The favorable spectral properties of the 3D bath of 1D tubes proposed in this paper might be advantageous also in this context.

## Acknowledgments

**Funding information** The work is funded by the Research Unit FOR 2414 of the Deutsche Forschungsgemeinschaft (DFG), Project No. 277974659. The work of C. W. is funded by the Cluster of Excellence "CUI: Advanced Imaging of Matter" of the DFG - EXC 2056 - Project ID No. 390715994 and by the European Research Council (ERC) under the European Union's Horizon 2020 research and innovation program under Grant Agreement No. 802701.

# A  High-frequency approximation for the system Hamiltonian

We start by introducing the annihilation operators $\hat{b}_{A/B}(\vec{k}) = (M_x M_y)^{-1/2} \sum_{\mathbf{l} \in A/B} \exp(i\vec{k}\vec{r}_{\mathbf{l}})\hat{a}_{\mathbf{l}}$ for at sublattice $A, B$ and quasimomentum $\hbar\vec{k}$ into the system Hamiltonian in Eq. (2)

$$\hat{H}_S(t) = -\sum_{\langle \mathbf{l}, \mathbf{l}' \rangle} J_n(t) \hat{a}_{\mathbf{l}}^\dagger \hat{a}_{\mathbf{l}'} = \sum_{\vec{k}} \left(\hat{b}_A^\dagger(\vec{k}), \hat{b}_B^\dagger(\vec{k})\right) \underbrace{\begin{pmatrix} 0 & h(\vec{k}, t) \\ h(\vec{k}, t)^* & 0 \end{pmatrix}}_{H_S(\vec{k}, t)} \begin{pmatrix} \hat{b}_A(\vec{k}) \\ \hat{b}_B(\vec{k}) \end{pmatrix}, \qquad \text{(A.1)}$$

with

$$h(\vec{k}, t) = -\sum_{n=1}^3 J_n(t) e^{i\vec{k}\vec{a}_n}. \qquad \text{(A.2)}$$

By using

$$J_n(t) = \frac{1}{2}\left(e^{A\cos(\omega t + \varphi_n)} + 1\right) = \frac{1}{2}\left(1 + \sum_{m \in \mathbb{Z}} I_m(A) e^{im\varphi_n} e^{im\omega t}\right), \qquad \text{(A.3)}$$

where $I_m(z)$ is the modified Bessel function of first kind, we can rewrite

$$H_S(\vec{k}, t) = -\sum_{m \in \mathbb{Z}} \left(h_m^x(\vec{k})\sigma_x + h_m^y(\vec{k})\sigma_y\right) e^{im\omega t} = \sum_{m \in \mathbb{Z}} H_m(\vec{k}) e^{im\omega t}, \qquad \text{(A.4)}$$

with

$$h_0^x(\vec{k}) = \frac{J}{2}(1 + I_0(A)) \sum_{n=1}^3 \cos(\vec{k}\vec{a}_n), \qquad \text{(A.5)}$$

$$h_0^y(\vec{k}) = -\frac{J}{2}(1 + I_0(A)) \sum_{n=1}^3 \sin(\vec{k}\vec{a}_n), \qquad \text{(A.6)}$$

$$h_{m \neq 0}^x(\vec{k}) = \frac{J}{2} I_m(A) \sum_{n=1}^3 e^{im\varphi_n} \cos(\vec{k}\vec{a}_n), \qquad \text{(A.7)}$$

$$h_{m \neq 0}^y(\vec{k}) = \frac{J}{2} I_m(A) \sum_{n=1}^3 e^{im\varphi_n} \sin(\vec{k}\vec{a}_n). \qquad \text{(A.8)}$$

In the two leading orders of the Magnus expansion, the Floquet Hamiltonian therefore reads

$$H_F(\vec{k}) = H_0(\vec{k}) + \sum_{m=1}^\infty \frac{[H_m(\vec{k}), H_{-m}(\vec{k})]}{m\hbar\omega} + \sum_{m \in \mathbb{Z} \backslash \{0\}} \frac{[H_0(\vec{k}), H_m(\vec{k})]}{m\hbar\omega}. \qquad \text{(A.9)}$$

By using standard Pauli matrix commutation relations we find

$$[H_m(\vec{k}), H_{-m}(\vec{k})] = 0, \qquad \text{(A.10)}$$

$$[H_0(\vec{k}), H_m(\vec{k})] = \sigma_z i \frac{J^2}{2}(1 + I_0(A))I_m(A) \sum_{i,j=1}^3 e^{im\varphi_j} \sin(\vec{k}(\vec{a}_i - \vec{a}_j)). \qquad \text{(A.11)}$$

Plugging this into Eq. (A.9) we find the Floquet Hamiltonian in the high-frequency expansion

$$\hat{H}_F = -\sum_{\vec{k}} \left(\hat{b}_A^\dagger(\vec{k}), \hat{b}_B^\dagger(\vec{k})\right) \begin{pmatrix} h_1(\vec{k}) & h_0(\vec{k}) \\ h_0(\vec{k})^* & -h_1(\vec{k}) \end{pmatrix} \begin{pmatrix} \hat{b}_A(\vec{k}) \\ \hat{b}_B(\vec{k}) \end{pmatrix}. \qquad \text{(A.12)}$$

On leading order $(1/\omega)^0$ we find the contribution

$$h_0(\vec{k}) = J_{\text{eff}}(A) \sum_n \exp(i\vec{k}\vec{a}_n), \tag{A.13}$$

with an effective uniform nearest-neighbor hopping $J_{\text{eff}}(A) = J(1 + I_0(A))/2$ where $I_m(z)$ is the modified Bessel function of first kind. The next order $(1/\omega)^1$ gives

$$h_1(\vec{k}) = J_{\text{eff}} J \sum_{m=1}^{\infty} I_m(A) \sum_{i,j=1}^{3} \sin(\vec{k}(\vec{a}_i - \vec{a}_j)) \frac{\sin(m\varphi_i)}{m\hbar\omega}, \tag{A.14}$$

where the dominant $m = 1$ term is remanent (but not exactly of the form) of the Haldane model. Calculating the Chern numbers for this Hamiltonian in Eq. (A.12) numerically, we find $C = -1, +1$.

# B Bogoliubov transformation for superfluid bath of 1D tubes

The microscopic bath Hamiltonian for an array of disconnected 1D tubes reads

$$\hat{H}_B = \sum_{\mathbf{l}} \int_z \hat{\chi}_{\mathbf{l}}^{\dagger}(z) \left( \frac{-\hbar^2}{2m_B} \frac{\partial^2}{\partial z^2} \right) \hat{\chi}_{\mathbf{l}}(z) + \frac{g}{2} \int_r \hat{\chi}^{\dagger}(\vec{r})\hat{\chi}^{\dagger}(\vec{r})\hat{\chi}(\vec{r})\hat{\chi}(\vec{r}), \tag{B.1}$$

where the index $\mathbf{l}$ is labeling all lattice sites of the two-dimensional honeycomb lattice of tubes and we use the convention $\int_r = \int d^3r$, $\int_z = \int dz$. The field operator for the bath particles can be expressed as

$$\hat{\chi}(\vec{r}) = \sum_{\mathbf{l}} w_{\mathbf{l}}^B(x, y)\hat{\chi}_{\mathbf{l}}(z), \tag{B.2}$$

with Wannier orbitals $w_{\mathbf{l}}^B$ of the bath at site $\mathbf{l}$. Neglecting contributions from Wannier orbitals at different lattice sites allows us to rewrite the Hamiltonian as

$$\hat{H}_B = \sum_{\mathbf{l}} \int_z \left\{ \hat{\chi}_{\mathbf{l}}^{\dagger}(z) \left( \frac{-\hbar^2}{2m_B} \frac{\partial^2}{\partial z^2} \right) \hat{\chi}_{\mathbf{l}}(z) + \frac{\tilde{g}}{2} \hat{\chi}_{\mathbf{l}}^{\dagger}(z)\hat{\chi}_{\mathbf{l}}^{\dagger}(z)\hat{\chi}_{\mathbf{l}}(z)\hat{\chi}_{\mathbf{l}}(z) \right\}, \tag{B.3}$$

with effective interaction strength $\tilde{g} = g \int_x \int_y |w_0^B(x, y)|^4$, where we use that the shape of the Wannier functions for both sub-lattices is identical up to a rotation.

To find an effective low-energy and low-temperature description of the bath Hamiltonian, we perform the usual semiclassical approximation that leads to a description in terms of Bogoliubov quasiparticles. After defining the momentum basis $\hat{c}_{\mathbf{l},q} = \frac{1}{\sqrt{L_z}} \int_z e^{-iqz} \hat{\chi}_{\mathbf{l}}(z)$ for temperatures $T$ well below $T_c^{\text{bath}}$ and weak interactions $g$ one may represent the bath field as $\hat{\chi}_{\mathbf{l}}(z) = \chi_{\mathbf{l},0} + \delta\hat{\chi}_{\mathbf{l}}(z)$. The first term is a c-number field $\chi_{\mathbf{l},0} = \sqrt{N_{\mathbf{l},0}/L_z} e^{i\varphi_{\mathbf{l}}}$ that describes the superfluid atoms. Here $\varphi_{\mathbf{l}}$ is the condensate phase at site $\mathbf{l}$ and occupations per tube $N_{\mathbf{l},0} = N_0/(2M_x M_y)$, i.e. we assume that $N_0$ bath bosons are equally distributed among the condensates in all the tubes. Additionally, there are small operator-valued fluctuations

$$\delta\hat{\chi}_{\mathbf{l}}(z) = \frac{1}{\sqrt{L_z}} \sum_{q \neq 0} e^{iqz} \hat{c}_{\mathbf{l},q}, \tag{B.4}$$

around it. Plugging the decomposition into the bath Hamiltonian, Eq. (B.3), we omit terms that are of higher order than $\delta\chi_1(z)^2$ to find

$$
\begin{aligned}
\hat{H}_{\mathrm{B}} \approx \sum_{1} \int_{z} \delta\hat{\chi}_1^\dagger(z)\left[\frac{-\hbar^2}{2m_{\mathrm{B}}}\frac{\partial^2}{\partial z^2}+G\right]\delta\hat{\chi}_1(z)+\frac{GN_{\mathrm{B}}}{2} \\
+\frac{G}{2}\sum_{1}\int_{z}\left[e^{-i2\varphi_1}\delta\hat{\chi}_1(z)\delta\hat{\chi}_1(z)+e^{i2\varphi_1}\delta\hat{\chi}_1^\dagger(z)\delta\hat{\chi}_1^\dagger(z)\right],
\end{aligned}
\tag{B.5}
$$

where we have used $N_B = \hat{N}_0 + \sum_1 \int_z \delta\hat{\chi}_1^\dagger(z)\delta\hat{\chi}_1(z)$ and introduced $G = \tilde{g}\tilde{n}_{\mathrm{B}}$ with tube density

$$
\tilde{n}_{\mathrm{B}} = \frac{N_{\mathrm{B}}}{2M_x M_y L_z}.
\tag{B.6}
$$

Alternatively, we may write $G = g n_{\mathrm{B}}$ with volume density

$$
n_{\mathrm{B}} = \tilde{n}_{\mathrm{B}}\int_{x}\int_{y}|w_0^{\mathrm{B}}(x,y)|^4.
\tag{B.7}
$$

We then use Eq. (B.4) and perform the standard Bogoliubov transformation

$$
\hat{\beta}_{1,q} = e^{-i\varphi_1}u_q\hat{c}_{1,q}+e^{-i\varphi_1}v_q\hat{c}_{1,-q}^\dagger,
\tag{B.8}
$$

to bring the Hamiltonian to the form

$$
\hat{H}_{\mathrm{B}} = \sum_{1}\sum_{q}E_{\mathrm{B}}(q)\hat{\beta}_{1,q}^\dagger\hat{\beta}_{1,q},
\tag{B.9}
$$

with Bogoliubov dispersion

$$
E_{\mathrm{B}}(q) = \sqrt{E_0(q)^2+2GE_0(q)},
\tag{B.10}
$$

where $E_0(q) = \hbar^2 q^2/2m_{\mathrm{B}}$, and the transformation follows from $u_q^2 - v_q^2 = 1$ and $u_q v_q = G/(2E_{\mathrm{B}}(q))$.

With this transformation, we may now rewrite the system-bath Hamiltonian as

$$
\hat{H}_{\mathrm{SB}} = \gamma\int_{r}\hat{\Psi}^\dagger(\vec{r})\hat{\Psi}(\vec{r})\hat{B}(\vec{r}),
\tag{B.11}
$$

with field operator $\hat{\Psi}(\vec{r})$ of the system and

$$
\hat{B}(\vec{r}) = \hat{\chi}^\dagger(\vec{r})\hat{\chi}(\vec{r})-\tilde{n}_{\mathrm{B}}\sum_{1}|w_1^{\mathrm{B}}(x,y)|^2
\tag{B.12}
$$

$$
\approx \sqrt{\tilde{n}_{\mathrm{B}}}\sum_{1}|w_1^{\mathrm{B}}(x,y)|^2\left[e^{-i\varphi_1}\delta\hat{\chi}_1(z)+e^{i\varphi_1}\delta\hat{\chi}_1^\dagger(z)\right]
\tag{B.13}
$$

$$
= \sqrt{\frac{\tilde{n}_{\mathrm{B}}}{L_z}}\sum_{1}|w_1^{\mathrm{B}}(x,y)|^2\sum_{q\neq 0}(u_q-v_q)\left[e^{iqz}\hat{\beta}_{1,q}+e^{-iqz}\hat{\beta}_{1,q}^\dagger\right].
\tag{B.14}
$$

In the second step we have we omitted terms that are of higher order in $\delta\chi_1(z)$, and in the last step we have employed Eq. (B.4), as well as the inverse Bogoliubov transformation $\hat{c}_{1,q} = e^{i\varphi_1}u_q\hat{\beta}_{1,q}-e^{i\varphi_1}v_q\hat{\beta}_{1,-q}^\dagger$.

For the system we rewrite the field operator using $\hat{\Psi}(\vec{r}) = \sum_{\mathbf{l}} w_{\mathbf{l}}^{S}(\vec{r}) a_{\mathbf{l}}$ with Wannier functions $w_{\mathbf{l}}^{S}(\vec{r})$, where the Wannier centers are at the same lattice sites as the bath. Thus, in leading order $\delta\hat{\chi}$, the system–bath coupling operator reads

$$\hat{H}_{SB} = \gamma \sum_{\mathbf{l},q\neq 0} \hat{n}_{\mathbf{l}} \left[ \kappa_{\mathbf{l}}(q)\hat{\beta}_{\mathbf{l},q} + \kappa_{\mathbf{l}}(q)^* \hat{\beta}_{\mathbf{l},q}^\dagger \right] = \gamma \sum_{\mathbf{l}} \hat{n}_{\mathbf{l}} \hat{B}_{\mathbf{l}}, \tag{B.15}$$

with coefficients

$$\kappa_{\mathbf{l}}(q) = \sqrt{\frac{\tilde{n}_B E_0(q)}{L_z E_B(q)}} \int_r |w_{\mathbf{l}}^{S}(\vec{r})|^2 |w_{\mathbf{l}}^{B}(x,y)|^2 e^{iqz}. \tag{B.16}$$

In the last step we again neglect all contributions from off-site Wannier orbitals. Finally, we can evaluate the $\kappa_{\mathbf{l}}(q)$ explicitly by approximating the Wannier functions with harmonic oscillator ground states

$$w_{\mathbf{l}}^{S}(\vec{r}) \approx \varphi_{S,L}^{HO}(x - x_{\mathbf{l}}) \varphi_{S,L}^{HO}(y - y_{\mathbf{l}}) \varphi_{S,T}^{HO}(z), \tag{B.17}$$

with effective frequency in the longitudinal directions of the lattice $\Omega_{S,L} = 2\sqrt{V_0 E_R}/\hbar$ and harmonic oscillator ground state $\varphi_{S,i}^{HO}(x) = (d_{S,i}\sqrt{\pi})^{-0.5} e^{-(x/d_{S,i})^2/2}$. Also $d_{S,i} = \sqrt{\hbar/m_S \Omega_{S,i}}$ denotes the harmonic oscillator length in the lattice ($i = L$) and the transverse ($i = T$) direction. Similarly, for the bath, we assume

$$w_{\mathbf{l}}^{B}(x,y) \approx \varphi_B^{HO}(x - x_{\mathbf{l}}) \varphi_B^{HO}(y - y_{\mathbf{l}}). \tag{B.18}$$

This yields

$$\kappa_{\mathbf{l}}(q) = \frac{1}{d} \sqrt{\frac{2 n_B E_0(q)}{\pi L_z E_B(q)}} e^{-\frac{1}{4} d_{S,T}^2 q^2}, \tag{B.19}$$

with length scale $d = d_B/(1 + d_B^2/d_{S,L}^2)$.

## C  Single-particle rates for a weakly interacting 1D Bose-condensed bath

Note that, omitting fluctuations of higher order than $\delta\hat{\chi}(\vec{r})$ (which means that we restrict ourself to one-phonon scattering in the bath, which largely dominates over higher-order phonon scattering for low temperatures $T$ [65]), $\hat{H}_{SB}$ is already in the desired form $\hat{H}_{SB} = \sum_i \hat{v}_i \otimes \hat{B}_i$ for the open quantum system formalism. The usual Born-, Markov- [106] and full rotating-wave approximation [61, 63, 107–110] lead to single-particle rates

$$R_{\alpha\beta} = \frac{2\pi\gamma^2}{\hbar} \text{Re} \sum_{K\in\mathbb{Z}} \sum_{\mathbf{l},\mathbf{l}'} (v_{\mathbf{l}})_{\alpha\beta}^{(K)*} (v_{\mathbf{l}'})_{\alpha\beta}^{(K)} W_{\mathbf{l}\mathbf{l}'}(\Delta_{\alpha\beta}^{(K)}), \tag{C.1}$$

describing a bath-induced quantum jump from Floquet state $\alpha$ to Floquet state $\beta$. Here we have defined the quasienergy difference $\Delta_{\alpha\beta}^{(K)} = \varepsilon_\alpha - \varepsilon_\beta + K\hbar\omega$, and the Fourier components of the coupling matrix

$$(v_{\mathbf{l}})_{\alpha\beta}^{(K)} = \frac{1}{\mathcal{T}} \int_0^{\mathcal{T}} dt\, e^{-iK\omega t} \langle u_\alpha(t)|\mathbf{l}\rangle\langle\mathbf{l}|u_\beta(t)\rangle = \sum_r u_{\alpha,\mathbf{l}}^{(r)*} u_{\beta,\mathbf{l}}^{(r+K)}, \tag{C.2}$$

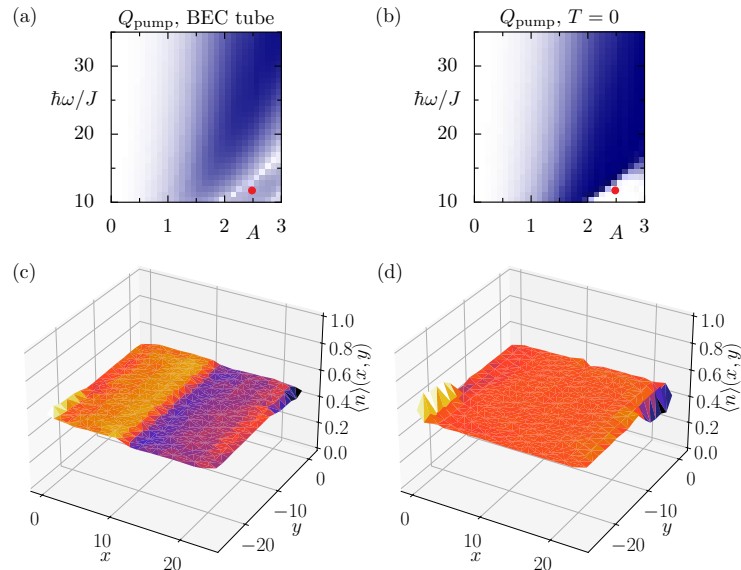

Figure 11: (a) Same as Fig. 7(c) and (b) same as Fig. 7(b), but marking the parameters $\hbar\omega = 11.7J, A = 2.5$ for the edge-mode dynamics in the anomalous Floquet topological insulator phase in (c) and (d). (c) Mean occupations $\langle n \rangle(x_i, y_i)$ at the lattice sites $(i, j)$ after the full flux insertion protocol of (a) with the bath of BEC tubes, but with $t_{\mathrm{p}} = 9.6\hbar/J_{\mathrm{eff}}(A)$. (d) Ideal edge-mode dynamics in the anomalous regime for the Floquet-Gibbs state at $T_{\mathrm{eff}} = 0$.

with driving period $\mathcal{T} = 2\pi/\omega$, and Floquet mode $|u_\alpha(t)\rangle$, as well as the $r$-th Fourier component $u_{\alpha,\mathbf{l}}^{(r)} = \langle \mathbf{l}|u_\alpha\rangle^{(r)}$ of Floquet mode $\alpha$. We have also employed the half-sided Fourier transform,

$$W_{\mathbf{l}\mathbf{l}'}(E) = \frac{1}{\pi\hbar} \int_0^\infty \mathrm{d}\tau \, \mathrm{e}^{-\frac{i}{\hbar}E\tau} \langle \tilde{B}_{\mathbf{l}}(\tau)\hat{B}_{\mathbf{l}'}\rangle_{\mathrm{B}}, \tag{C.3}$$

of the bath correlation function, we denote $\langle \cdot \rangle_{\mathrm{B}} = \mathrm{Tr}_{\mathrm{B}}\hat{\varrho}_{\mathrm{B}}\cdot$ and the tilde in $\tilde{B}_i(\tau)$ indicates the operator in the interaction picture, where for a general operator

$$\tilde{O}(\tau) = \mathrm{e}^{i(\hat{H}_{\mathrm{S}}+\hat{H}_{\mathrm{B}})\tau}\hat{O}\mathrm{e}^{-i(\hat{H}_{\mathrm{S}}+\hat{H}_{\mathrm{B}})\tau}. \tag{C.4}$$

We use that the bath is in a thermal state $\hat{\varrho}_{\mathrm{B}} = \frac{1}{Z}\exp(-\hat{H}_{\mathrm{B}}/k_{\mathrm{B}}T)$, to evaluate

$$\langle \tilde{B}_{\mathbf{l}}(t)\hat{B}_{\mathbf{l}'}\rangle_{\mathrm{B}} = \sum_{q,q'\neq 0} \left\langle \left[ \kappa_{\mathbf{l}}(q)\hat{\beta}_{\mathbf{l},q}\mathrm{e}^{-\frac{i}{\hbar}E_{\mathrm{B}}(q)t} + \kappa_{\mathbf{l}}(q)^*\hat{\beta}_{\mathbf{l},q}^\dagger\mathrm{e}^{\frac{i}{\hbar}E_{\mathrm{B}}(q)t} \right]\left[ \kappa_{\mathbf{l}'}(q')\hat{\beta}_{\mathbf{l}',q'} + \kappa_{\mathbf{l}'}(q')^*\hat{\beta}_{\mathbf{l}',q'}^\dagger \right] \right\rangle_{\mathrm{B}} \tag{C.5}$$

$$= \delta_{\mathbf{l}\mathbf{l}'} \int_{-\infty}^\infty \mathrm{d}E \, \mathcal{J}(E)\mathrm{e}^{\frac{i}{\hbar}Et}n(E), \tag{C.6}$$

with Bose-Einstein occupation function of the bath

$$n(E) = \frac{1}{\mathrm{e}^{E/k_B T} - 1}, \tag{C.7}$$

and spectral density

$$\mathcal{J}(E) = \sum_{q\neq 0} |\kappa_{\mathbf{l}}(q)|^2 \left[ \delta(E - E_{\mathrm{B}}(q)) - \delta(E + E_{\mathrm{B}}(q)) \right], \tag{C.8}$$

where we use that $\kappa_1(q)$ is independent of $\mathbf{l}$. Therefore, using the Sokhotski-Plemelj formula gives

$$W_{\mathbf{ll'}}(E) = \delta_{\mathbf{ll'}} \mathcal{J}(E) n(E). \tag{C.9}$$

Finally, we take the continuum limit for the bath sum over $q$, $\frac{2\pi}{L_z} \sum_q \to \int dq$, to obtain

$$\mathcal{J}(E) = \frac{n_B}{d^2 \pi^2} \int dq \frac{E_0(q)}{E_B(q)} e^{-\frac{1}{2} d_{T,s}^2 q^2} \delta(E - E_B(q)), \tag{C.10}$$

for $E > 0$ and $\mathcal{J}(-E) = -\mathcal{J}(E)$. We solve Eq. (B.10) for the momentum

$$q(E) = \frac{\sqrt{2m_B}}{\hbar} \left( \sqrt{E^2 + G^2} - G \right)^{1/2}, \tag{C.11}$$

of a Bogoliubov quasiparticle at Energy $E$, so that

$$2q dq = \frac{2m_B}{\hbar^2} \frac{E_B}{\sqrt{E_B^2 + G^2}} dE_B. \tag{C.12}$$

After transforming the $q$-integral into an integral over $E_B$, we can directly evaluate the delta distribution and find

$$\mathcal{J}(E) = \text{sgn}(E) \frac{n_B}{d^2 \pi^2 2} \frac{q(E)}{\sqrt{E^2 + G^2}} e^{-\frac{1}{2} d_{T,s}^2 q(E)^2}. \tag{C.13}$$

Note that for small energies $E \ll G$ we find sub-ohmic behavior $\mathcal{J}(E) \propto E^{1/2}$, while for $E \gg G$ the spectral density decays again exponentially.

## D  Edge mode dynamics

In Fig. 11(c),(d) we show a snapshot of the real-space density $\langle n \rangle(x, y)$ at $t = t_p$, after the Laughlin-type charge pumping in the anomalous Floquet topological insulator phase for the parameters of the red dot in Fig. 11(a),(b). Fig. 11(c) corresponds to the BEC tube bath, and Fig. 11(d) to the ideal Floquet-Gibbs state at effective temperature $T_{\text{eff}} = 0$. We observe that, due to the high effective temperature at this parameter set, the edge modes are much less visible in case of the engineered bath of 1D tubes when compared to the ideal case. Nonetheless, there is indication of the topologically nontrivial behavior in the form of edge modes in the anomalous Floquet phase (however at a reduced contrast). In order to display the edge mode more clearly, we have increased the pumping time to $t_p = 9.6\hbar/J_{\text{eff}}(A)$.

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
