# Peer review of "Dissipative preparation of a Floquet topological insulator in an optical lattice via bath engineering"

_SciPost Physics, doi:SciPost Phys. 17, 052 (2024)_

## Round 2 · Referee Report · Anonymous (Referee 1) · 2023-8-9

Strengths
1- the paper proposes a solution to the outstanding challenge of preparing a genuine 2D topological insulator in optical lattices 2- the proposal uses a combination of existing experimental techniques, including a possible observable for the topological insulator (quantised pumping) 3- it compares concrete alternatives (different combinations of species and experimental settings) and suggests an optimal configuration (large mass imbalance, 1D bath tubes)
Weaknesses
1- the required physical ingredients of the proposal lead to a high overall experimental complexity 2- there are some open questions regarding the ratio of polarisabilities for the two atomic species 3- the method for detecting the topological insulator (quantised pumping) appears a bit convoluted
Report
The authors elegantly solve both problems by engineering a very specific bath configuration, namely one-dimensional tubes along the orthogonal direction of the 2D topological-insulator lattice. These tubes contain heavy bosonic atoms, which interact weakly with the lighter fermions of the lattice.
The density-of-states of the bosons must be suitably narrow in energy, ideally smaller than the drive frequency. In this configuration, the fermions automatically form a near-perfect topological insulator as the non-equilibrium steady state.
In general, the manuscript meets all the necessary criteria for SciPost Physics (well-written, contains abstract, citations, specific details of derivations). Furthermore, it addresses an important, currently unsolved problem in the field of quantum simulation of topological matter. Therefore, the paper has the potential to be published in SciPost Physics, following some minor revisions.
Requested changes
1- The authors quickly converge towards establishing the mass imbalance of the two species as relevant parameter for successful bath engineering. While a large momentum transfer in system-bath collisions certainly makes sense intuitively, the aspect of lattice depths due to different polarisabilities (how near-resonant the lattice is) for the two species remains largely unexplored. As mentioned by the authors in the context of tube-versus-2D confinement, it is possible to find tune-out wavelengths in which one species does not feel the confinement of the other species.
I think the authors may have overlooked this aspect in terms of the main lattice potential. As I understand it, the most important criterion for the bath engineering is that the tubes of bosons are very disconnected, while the lattice remains dispersive for the fermions. The disconnected nature of tubes leads to the correct density-of-states in the bath (Fig. 5a).
Here, I would suggest taking a closer look at the lattice wavelength with respect to the atomic transitions. The proposed optimal scenario (lithium-6, cesium-133, lambda = 740nm) may turn out to be unrealistic, unless I have overlooked something, because the wavelength 740nm is mainly repulsive for cesium (D-lines around 850-890nm) while it is attractive for lithium-6 (D-lines at 671nm). In this case, the bath tubes would lie on the bonds of the lattice, not at the lattice sites.
2- Verifying a 2D topological insulator experimentally is in itself quite a challenging task. When measuring a quantised bulk-Hall response in finite (box) system, for instance, one would find an accumulation of charge at the boundary and then a reflection of the quantised current via edge states and the upper band with C = -1.
The difficulty of detection is apparent, as even in the ideal (T = 0) system the response is only ~90% 'quantised' (Fig. 7a). Thus, the chosen method for detecting the topological insulator appears sub-optimal.
In addition, I would like to see how the Peierls phases are generated concretely. I don't yet understand how adding a step-potential leads to the stripe of Peierls phases. Traditionally, I would have expected typical quantum Hall response, for example, by switching on a linear gradient along the x-direction leading to a quantised bulk current along y.
Evaluating the current operator across a single bond may provide another observable, instead of summing the entire charge in the upper half of the system.
Referee 1:
Strengths 1- the paper proposes a solution to the outstanding challenge of preparing a genuine 2D topological insulator in optical lattices 2- the proposal uses a combination of existing experimental techniques, including a possible observable for the topological insulator (quantised pumping) 3- it compares concrete alternatives (different combinations of species and experimental settings) and suggests an optimal configuration (large mass imbalance, 1D bath tubes)
Weaknesses 1- the required physical ingredients of the proposal lead to a high overall experimental complexity 2- there are some open questions regarding the ratio of polarisabilities for the two atomic species 3- the method for detecting the topological insulator (quantised pumping) appears a bit convoluted
Report The manuscript tackles one of the outstanding challenges in the field of quantum simulation, namely the preparation of a genuine 2D topological insulator. Two main challenges must be met to prepare a 2D topological insulator. On the one hand, a band gap must necessarily close during the transition from trivial to topological band. On the other hand, the Floquet driving, necessary for creating topological bands, can lead to unwanted heating out of the topologically insulating state. The authors elegantly solve both problems by engineering a very specific bath configuration, namely one-dimensional tubes along the orthogonal direction of the 2D topological-insulator lattice. These tubes contain heavy bosonic atoms, which interact weakly with the lighter fermions of the lattice. The density-of-states of the bosons must be suitably narrow in energy, ideally smaller than the drive frequency. In this configuration, the fermions automatically form a near-perfect topological insulator as the non-equilibrium steady state. In general, the manuscript meets all the necessary criteria for SciPost Physics (well-written, contains abstract, citations, specific details of derivations). Furthermore, it addresses an important, currently unsolved problem in the field of quantum simulation of topological matter. Therefore, the paper has the potential to be published in SciPost Physics, following some minor revisions.
Our response:
We thank the referee for the very detailed and constructive evaluation of our manuscript.
Referee 1:
Requested changes 1- The authors quickly converge towards establishing the mass imbalance of the two species as relevant parameter for successful bath engineering. While a large momentum transfer in system-bath collisions certainly makes sense intuitively, the aspect of lattice depths due to different polarisabilities (how near-resonant the lattice is) for the two species remains largely unexplored. As mentioned by the authors in the context of tube-versus-2D confinement, it is possible to find tune-out wavelengths in which one species does not feel the confinement of the other species. I think the authors may have overlooked this aspect in terms of the main lattice potential. As I understand it, the most important criterion for the bath engineering is that the tubes of bosons are very disconnected, while the lattice remains dispersive for the fermions. The disconnected nature of tubes leads to the correct density-of-states in the bath (Fig. 5a). Here, I would suggest taking a closer look at the lattice wavelength with respect to the atomic transitions. The proposed optimal scenario (lithium-6, cesium-133, lambda = 740nm) may turn out to be unrealistic, unless I have overlooked something, because the wavelength 740nm is mainly repulsive for cesium (D-lines around 850-890nm) while it is attractive for lithium-6 (D-lines at 671nm). In this case, the bath tubes would lie on the bonds of the lattice, not at the lattice sites.
Our response:
We thank the referee for pointing out this inconsistency. We agree that for 740nm the lattice would be blue-detuned for Cs, therefore the lattice minima for system and bath would not coincide. In the revised manuscript we therefore have changed the wavelength of the lattice to lambda=1064nm. For Cs, the detuning would be then less than for Li, which is in line with our assumption of a deeper lattice for the bath.
We have modified all the respective plots in the manuscript for the new parameter lambda=1064nm and now write on page 3:
„Additionally, since the different atomic species possess different polarizabilities, also the lattice depth $V_0$ will be different for each species. Later we propose a choice of an optical wavelength that leads to larger values of $V_0$ for the bath when compared to the system.“
And on page 6:
„For the hexagonal lattice we assume an optical wavelength $\lambda = 1064 \mathrm{nm}$, which is red-detuned for both atomic species. However, the detuning is larger for $^6$Li than for $^{133}$Cs, which leads to a deeper lattice for the bath atoms as we imagine.“
Referee 1: 2- Verifying a 2D topological insulator experimentally is in itself quite a challenging task. When measuring a quantised bulk-Hall response in finite (box) system, for instance, one would find an accumulation of charge at the boundary and then a reflection of the quantised current via edge states and the upper band with C = -1.
Our response:
This is exactly what happens for our protocol and limits the time scale of flux insertion in the pumping protocol as we discuss around Eq. (27).
Referee 1: The difficulty of detection is apparent, as even in the ideal (T = 0) system the response is only ~90% 'quantised' (Fig. 7a). Thus, the chosen method for detecting the topological insulator appears sub-optimal. In addition, I would like to see how the Peierls phases are generated concretely. I don't yet understand how adding a step-potential leads to the stripe of Peierls phases. Traditionally, I would have expected typical quantum Hall response, for example, by switching on a linear gradient along the x-direction leading to a quantised bulk current along y. Evaluating the current operator across a single bond may provide another observable, instead of summing the entire charge in the upper half of the system.
Our response:
We agree that in the original version of the manuscript the insertion of the Peirls phases was discussed too vaguely, so we have now added a paragraph on Page 7:
„This linear increase of the Peierls phase along the red links can be induced easily by applying an additional on-site potential δ = −2πħ/tp on all lattice sites right of the red colored links in Fig. 2, described by the Hamiltonian …“
which describes how this can be achieved by an addition of a constant on-site potentials on the right half of the system.
Regarding the non-ideal pumping scheme, we have optimized our scheme in two ways to address the concerns of the referee: 1) We have optimized our simulations and are now able to host 16x16 unit cells, reducing finite-size effects. 2) We have adapted the pumping time for the effective tunneling rate that we obtain from the high-frequency expansion.
With that we find a large parameter regime (with $A \gtrsim 2.5$) in which there is very close to unit charge pumping in the ideal case of T=0, and an almost unit charge pumping in presence of the engineered bath.
Referee 1:
Validity: Good
Significance: Top
Originality: Top
Clarity: High
Formatting: Excellent
Grammar: Good

Author: Alexander Schnell on 2024-06-07 [id 4545]
(in reply to Report 2 on 2023-09-11)Referee 2:
Report
In this manuscript, the authors study periodically driven fermions in an optical lattice coupled to an engineered bosonic bath. This setup is motivated by quantum-gas experiments with a Bose-Fermi mixture, where bosonic atoms can act as a bath for fermionic atoms. Specifically, they consider a situation where fermions are confined in a two-dimensional honeycomb lattice and bosons form quasi-1D Bose-Einstein condensates (BECs) at each lattice site. While the fermions are heated up by periodic driving, this heating is suppressed by the coupling to the bosonic bath. The authors find that the cooling is efficient due to suppression of the spectral density of the bath at high energies if one uses quasi-1D BECs as the bath and the mass of bosonic atoms is sufficiently larger than that of fermionic atoms. As a result, Floquet topological insulator phases of fermions are stabilized and show quantized responses, which are confirmed by numerical simulations.
Floquet engineering is now a useful tool in quantum-gas experiments, but the associated heating is often problematic. The proposal in this manuscript is interesting and appealing since they consider a concrete and realistic setting for the implementation of a bath with bosonic atoms such as 133Cs. Below I enclose my questions and comments on the manuscript:
Our response:
We thank the referee for the very detailed and constructive evaluation of our manuscript.
Referee 2:
(1) In Fig. 4(a), the authors show the Chern number of the lowest Floquet band, but the system size is small (it contains only 16 unit cells). How did the authors extract the Chern number from such small systems? Is there any finite-size effect on this calculation?
Our response:
We have used the procedure that is outlined in Eq. (20) of Ref. [Opt. Express 28, 4638 (2020)], which we now specify on page 4: „In Fig. 4(a) we show the Chern number C that we obtain numerically (according to Ref. [105]) for the lower quasienergy band… “. To make sure that there are no finite size effects, in the revised version we have calculated Fig. 4 for the case of 16x16 unit cells.
Referee 2:
(2) The authors claim that the anomalous Floquet topological insulator is also stabilized by this scheme. However, the authors only show that the pumped charge drops to zero in the anomalous Floquet topological phase. Since a trivial insulator also shows zero pumped charge, it is not enough for claiming that this is an anomalous Floquet topological insulator. Can the authors show a decisive signature of this anomalous topological phase, e.g., the existence of edge states, in this dissipation-engineered Floquet system?
Our response:
Throughout the manuscript, we stress that the transition to the anomalous Floquet TI phase is only observed with a weak indication. Our stabilization protocol generally works well only in the Haldane-like phase. Nevertheless, there is visible edge-mode dynamics at the end of the pumping cycle also in the anomalous regime for the BEC tube bath as we now show in Appendix D, Fig.11. Due to the high effective temperature, however, the amplitude of the edge mode is significantly reduced when compared to the ideal T_eff=0 case.
On page 8, we now write:
„However, the thin white stripe in Fig. 7(c) suggests that there is a small parameter regime, where Umklapp processes inside the system can already give rise to the anomalous Floquet topological band structure, while the Umklapp processes associated with the system-bath coupling are not yet detrimental for the preparation of an approximate band insulator. This is substantiated by Appendix D, where we show visible edge mode dynamics in the anomalous Floquet topological insulator phase in the steady state of the system with the BEC tube bath, albeit at reduced contrast when compared to the ideal T_eff = 0 case.“
Referee 2:
(3) The effective temperature in Fig. 6(c) shows that the cooling seems inefficient in the anomalous Floquet topological phase. Can the authors explain the origin of this behavior? Is it related to the gap closing associated with the topological phase transition?
Our response:
As the referee correctly points out, this behavior is due to the gap closing associated with the topological phase transition, leading to processes from the Floquet copy of the highly occupied lower band to the upper band. These processes are not blocked by the spectral density, since a non-vanishing spectral density around E=0 is needed to allow for thermalization in the Haldane phase. We have added a sentence on page 8 to make this point more clear:
„At the topological phase transition, there is a closing of the gap across the first and second Floquet-Brillouin zone, and the bath allows for population transfer from the Floquet copy of the lower Floquet band to the upper Floquet band (cf. processes with rate R(−1) in Fig. 3).“
Referee 2:
(4) In Sec. II A, the authors write that the model is described by a Hubbard-Holstein Hamiltonian. This is a little misleading since the system is non-interacting fermions and does not have a Hubbard interaction.
Our response:
We have clarified that point on page 3, by removing „Hubbard“ and adding to the sentence „this model is described by a Holstein Hamiltonian (with vanishing interactions in the system, cf. Appendix B and Refs. [64, 97])“
Referee 2:
(5) In Eq. (7), the system-bath Hamiltonian is time-dependent, while the right-hand side appears to be time-independent.
Our response:
We agree with the referee. The system-bath Hamiltonian is time-independent. We have removed the time-dependence on the left hand side.
Referee 2:
(6) Below Eq. (22), the authors write "with the lattice-trapped bath (red line), with the ohmic bath (green line)". It seems that the colors do not correspond to those in Fig. 7(a).
Our response:
We thank the referee for pointing us to that error. We have replaced the text with the correct referencing of the colors.
Referee 2:
In conclusion, this manuscript shows a promising way to stabilizing Floquet topological phases and provides a guide for experiments in the near future.
Validity: High
Significance: Good
Originality: Good
Clarity: Ok
Formatting: Good
Grammar: Excellent

---

## Round 2 · Referee Report · Anonymous (Referee 2) · 2023-9-11

Report
Floquet engineering is now a useful tool in quantum-gas experiments, but the associated heating is often problematic. The proposal in this manuscript is interesting and appealing since they consider a concrete and realistic setting for the implementation of a bath with bosonic atoms such as 133Cs. Below I enclose my questions and comments on the manuscript:
(1) In Fig. 4(a), the authors show the Chern number of the lowest Floquet band, but the system size is small (it contains only 16 unit cells). How did the authors extract the Chern number from such small systems? Is there any finite-size effect on this calculation?
(2) The authors claim that the anomalous Floquet topological insulator is also stabilized by this scheme. However, the authors only show that the pumped charge drops to zero in the anomalous Floquet topological phase. Since a trivial insulator also shows zero pumped charge, it is not enough for claiming that this is an anomalous Floquet topological insulator. Can the authors show a decisive signature of this anomalous topological phase, e.g., the existence of edge states, in this dissipation-engineered Floquet system?
(3) The effective temperature in Fig. 6(c) shows that the cooling seems inefficient in the anomalous Floquet topological phase. Can the authors explain the origin of this behavior? Is it related to the gap closing associated with the topological phase transition?
(4) In Sec. II A, the authors write that the model is described by a Hubbard-Holstein Hamiltonian. This is a little misleading since the system is non-interacting fermions and does not have a Hubbard interaction.
(5) In Eq. (7), the system-bath Hamiltonian is time-dependent, while the right-hand side appears to be time-independent.
(6) Below Eq. (22), the authors write "with the lattice-trapped bath (red line), with the ohmic bath (green line)". It seems that the colors do not correspond to those in Fig. 7(a).
In conclusion, this manuscript shows a promising way to stabilizing Floquet topological phases and provides a guide for experiments in the near future.

---

## Round 3 · Referee Report · Anonymous (Referee 1) · 2024-6-11

Report

The authors have satisfactorily addressed all concerns raised by the referees. I support publication in SciPost Physics.

Recommendation

Publish (surpasses expectations and criteria for this Journal; among top 10%)

---

## Round 3 · Referee Report · Anonymous (Referee 2) · 2024-6-21

Report

The authors have made appropriate changes in response to the questions by the referees. Now I believe that this manuscript is ready for publication.

Recommendation

Publish (easily meets expectations and criteria for this Journal; among top 50%)

---

## Round 3 · Author Response

Referee 1:

Strengths 1- the paper proposes a solution to the outstanding challenge of preparing a genuine 2D topological insulator in optical lattices
 2- the proposal uses a combination of existing experimental techniques, including a possible observable for the topological insulator (quantised pumping)
 3- it compares concrete alternatives (different combinations of species and experimental settings) and suggests an optimal configuration (large mass imbalance, 1D bath tubes) Weaknesses 1- the required physical ingredients of the proposal lead to a high overall experimental complexity
 2- there are some open questions regarding the ratio of polarisabilities for the two atomic species
 3- the method for detecting the topological insulator (quantised pumping) appears a bit convoluted Report The manuscript tackles one of the outstanding challenges in the field of quantum simulation, namely the preparation of a genuine 2D topological insulator. Two main challenges must be met to prepare a 2D topological insulator. On the one hand, a band gap must necessarily close during the transition from trivial to topological band. On the other hand, the Floquet driving, necessary for creating topological bands, can lead to unwanted heating out of the topologically insulating state. The authors elegantly solve both problems by engineering a very specific bath configuration, namely one-dimensional tubes along the orthogonal direction of the 2D topological-insulator lattice. These tubes contain heavy bosonic atoms, which interact weakly with the lighter fermions of the lattice. The density-of-states of the bosons must be suitably narrow in energy, ideally smaller than the drive frequency. In this configuration, the fermions automatically form a near-perfect topological insulator as the non-equilibrium steady state. In general, the manuscript meets all the necessary criteria for SciPost Physics (well-written, contains abstract, citations, specific details of derivations). Furthermore, it addresses an important, currently unsolved problem in the field of quantum simulation of topological matter. Therefore, the paper has the potential to be published in SciPost Physics, following some minor revisions.

Our response:

We thank the referee for the very detailed and constructive evaluation of our manuscript.

Referee 1:

Requested changes 1- The authors quickly converge towards establishing the mass imbalance of the two species as relevant parameter for successful bath engineering. While a large momentum transfer in system-bath collisions certainly makes sense intuitively, the aspect of lattice depths due to different polarisabilities (how near-resonant the lattice is) for the two species remains largely unexplored. As mentioned by the authors in the context of tube-versus-2D confinement, it is possible to find tune-out wavelengths in which one species does not feel the confinement of the other species.

I think the authors may have overlooked this aspect in terms of the main lattice potential. As I understand it, the most important criterion for the bath engineering is that the tubes of bosons are very disconnected, while the lattice remains dispersive for the fermions. The disconnected nature of tubes leads to the correct density-of-states in the bath (Fig. 5a). 

Here, I would suggest taking a closer look at the lattice wavelength with respect to the atomic transitions. The proposed optimal scenario (lithium-6, cesium-133, lambda = 740nm) may turn out to be unrealistic, unless I have overlooked something, because the wavelength 740nm is mainly repulsive for cesium (D-lines around 850-890nm) while it is attractive for lithium-6 (D-lines at 671nm). In this case, the bath tubes would lie on the bonds of the lattice, not at the lattice sites.

Our response:

We thank the referee for pointing out this inconsistency. We agree that for 740nm the lattice would be blue-detuned for Cs, therefore the lattice minima for system and bath would not coincide. In the revised manuscript we therefore have changed the wavelength of the lattice to lambda=1064nm. For Cs, the detuning would be then less than for Li, which is in line with our assumption of a deeper lattice for the bath.

We have modified all the respective plots in the manuscript for the new parameter lambda=1064nm and now write on page 3:

„Additionally, since the different atomic species possess different polarizabilities, also the lattice depth $V_0$ will be different for each species. Later we propose a choice of an optical wavelength that leads to larger values of $V_0$ for the bath when compared to the system.“

And on page 6:

„For the hexagonal lattice we assume an optical wavelength $\lambda = 1064 \mathrm{nm}$, which is red-detuned for both atomic species. However, the detuning is larger for $^6$Li than for $^{133}$Cs, which leads to a deeper lattice for the bath atoms as we imagine.“

Referee 1:

2- Verifying a 2D topological insulator experimentally is in itself quite a challenging task. When measuring a quantised bulk-Hall response in finite (box) system, for instance, one would find an accumulation of charge at the boundary and then a reflection of the quantised current via edge states and the upper band with C = -1.

Our response:

This is exactly what happens for our protocol and limits the time scale of flux insertion in the pumping protocol as we discuss around Eq. (27).

Referee 1:

The difficulty of detection is apparent, as even in the ideal (T = 0) system the response is only ~90% 'quantised' (Fig. 7a). Thus, the chosen method for detecting the topological insulator appears sub-optimal. 

In addition, I would like to see how the Peierls phases are generated concretely. I don't yet understand how adding a step-potential leads to the stripe of Peierls phases. Traditionally, I would have expected typical quantum Hall response, for example, by switching on a linear gradient along the x-direction leading to a quantised bulk current along y.

Evaluating the current operator across a single bond may provide another observable, instead of summing the entire charge in the upper half of the system.

Our response:

We agree that in the original version of the manuscript the insertion of the Peirls phases was discussed too vaguely, so we have now added a paragraph on Page 7:

„This linear increase of the Peierls phase along the red links can be induced easily by applying an additional on-site potential δ = −2πħ/tp on all lattice sites right of the red colored links in Fig. 2, described by the Hamiltonian …“

which describes how this can be achieved by an addition of a constant on-site potentials on the right half of the system.

Regarding the non-ideal pumping scheme, we have optimized our scheme in two ways to address the concerns of the referee: 1) We have optimized our simulations and are now able to host 16x16 unit cells, reducing finite-size effects. 2) We have adapted the pumping time for the effective tunneling rate that we obtain from the high-frequency expansion.

With that we find a large parameter regime (with A \gtrsim 2.5) in which there is very close to unit charge pumping in the ideal case of T=0, and an almost unit charge pumping in presence of the engineered bath.

Referee 1:

    Validity: Good
    Significance: Top
    Originality: Top
    Clarity: High
    Formatting: Excellent
    Grammar: Good

Referee 2:

Report In this manuscript, the authors study periodically driven fermions in an optical lattice coupled to an engineered bosonic bath. This setup is motivated by quantum-gas experiments with a Bose-Fermi mixture, where bosonic atoms can act as a bath for fermionic atoms. Specifically, they consider a situation where fermions are confined in a two-dimensional honeycomb lattice and bosons form quasi-1D Bose-Einstein condensates (BECs) at each lattice site. While the fermions are heated up by periodic driving, this heating is suppressed by the coupling to the bosonic bath. The authors find that the cooling is efficient due to suppression of the spectral density of the bath at high energies if one uses quasi-1D BECs as the bath and the mass of bosonic atoms is sufficiently larger than that of fermionic atoms. As a result, Floquet topological insulator phases of fermions are stabilized and show quantized responses, which are confirmed by numerical simulations. Floquet engineering is now a useful tool in quantum-gas experiments, but the associated heating is often problematic. The proposal in this manuscript is interesting and appealing since they consider a concrete and realistic setting for the implementation of a bath with bosonic atoms such as 133Cs. Below I enclose my questions and comments on the manuscript:

Our response:

We thank the referee for the very detailed and constructive evaluation of our manuscript.

Referee 2:

(1) In Fig. 4(a), the authors show the Chern number of the lowest Floquet band, but the system size is small (it contains only 16 unit cells). How did the authors extract the Chern number from such small systems? Is there any finite-size effect on this calculation?

Our response:

We have used the procedure that is outlined in Eq. (20) of Ref. [Opt. Express 28, 4638 (2020)], which we now specify on page 4: „In Fig. 4(a) we show the Chern number C that we obtain numerically (according to Ref. [105]) for the lower quasienergy band… “. To make sure that there are no finite size effects, in the revised version we have calculated Fig. 4 for the case of 16x16 unit cells.

Referee 2:

(2) The authors claim that the anomalous Floquet topological insulator is also stabilized by this scheme. However, the authors only show that the pumped charge drops to zero in the anomalous Floquet topological phase. Since a trivial insulator also shows zero pumped charge, it is not enough for claiming that this is an anomalous Floquet topological insulator. Can the authors show a decisive signature of this anomalous topological phase, e.g., the existence of edge states, in this dissipation-engineered Floquet system?

Our response:

Throughout the manuscript, we stress that the transition to the anomalous Floquet TI phase is only observed with a weak indication. Our stabilization protocol generally works well only in the Haldane-like phase. Nevertheless, there is visible edge-mode dynamics at the end of the pumping cycle also in the anomalous regime for the BEC tube bath as we now show in Appendix D, Fig.11. Due to the high effective temperature, however, the amplitude of the edge mode is significantly reduced when compared to the ideal T_eff=0 case.

On page 8, we now write:

„However, the thin white stripe in Fig. 7(c) suggests that there is a small parameter regime, where Umklapp processes inside the system can already give rise to the anomalous Floquet topological band structure, while the Umklapp processes associated with the system-bath coupling are not yet detrimental for the preparation of an approximate band insulator. This is substantiated by Appendix D, where we show visible edge mode dynamics in the anomalous Floquet topological insulator phase in the steady state of the system with the BEC tube bath, albeit at reduced contrast when compared to the ideal T_eff = 0 case.“

Referee 2:

(3) The effective temperature in Fig. 6(c) shows that the cooling seems inefficient in the anomalous Floquet topological phase. Can the authors explain the origin of this behavior? Is it related to the gap closing associated with the topological phase transition?

Our response:

As the referee correctly points out, this behavior is due to the gap closing associated with the topological phase transition, leading to processes from the Floquet copy of the highly occupied lower band to the upper band. These processes are not blocked by the spectral density, since a non-vanishing spectral density around E=0 is needed to allow for thermalization in the Haldane phase. We have added a sentence on page 8 to make this point more clear:

„At the topological phase transition, there is a closing of the gap across the first and second Floquet-Brillouin zone, and the bath allows for population transfer from the Floquet copy of the lower Floquet band to the upper Floquet band (cf. processes with rate R(−1) in Fig. 3).“

Referee 2:

(4) In Sec. II A, the authors write that the model is described by a Hubbard-Holstein Hamiltonian. This is a little misleading since the system is non-interacting fermions and does not have a Hubbard interaction.

Our response:

We have clarified that point on page 3, by removing „Hubbard“ and adding to the sentence „this model is described by a Holstein Hamiltonian (with vanishing interactions in the system, cf. Appendix B and Refs. [64, 97])“

Referee 2:

(5) In Eq. (7), the system-bath Hamiltonian is time-dependent, while the right-hand side appears to be time-independent.

Our response:

We agree with the referee. The system-bath Hamiltonian is time-independent. We have removed the time-dependence on the left hand side.

Referee 2:

(6) Below Eq. (22), the authors write "with the lattice-trapped bath (red line), with the ohmic bath (green line)". It seems that the colors do not correspond to those in Fig. 7(a).

Our response:

We thank the referee for pointing us to that error. We have replaced the text with the correct referencing of the colors.

Referee 2:

In conclusion, this manuscript shows a promising way to stabilizing Floquet topological phases and provides a guide for experiments in the near future.

    Validity: High
    Significance: Good
    Originality: Good
    Clarity: Ok
    Formatting: Good
    Grammar: Excellent

---

## Round 3 · List of Changes

See Author comments.

---

## Editorial Decision

published